# Learning Joint Wasserstein Auto-Encoders for Joint Distribution Matching

## Abstract

We study the joint distribution matching problem which aims at learning bidirectional mappings to match the joint distribution of two domains. This problem occurs in unsupervised image-to-image translation and video-to-video synthesis tasks, which, however, has two critical challenges: (i) it is difficult to exploit sufficient information from the joint distribution; (ii) how to theoretically and experimentally evaluate the generalization performance remains an open question. To address the above challenges, we propose a new optimization problem and design a novel Joint Wasserstein Auto-Encoders (JWAE) to minimize the Wasserstein distance of the joint distributions in two domains. We theoretically prove that the generalization ability of the proposed method can be guaranteed by minimizing the Wasserstein distance of joint distributions. To verify the generalization ability, we apply our method to unsupervised video-to-video synthesis by performing video frame interpolation and producing visually smooth videos in two domains, simultaneously. Both qualitative and quantitative comparisons demonstrate the superiority of our method over several state-of-the-arts.

## 1 Introduction

The joint distribution matching problem has attracted extensive attention in computer vision, such as unsupervised image-to-image translation (Zhu et al., 2017; Liu et al., 2017) and video-to-video synthesis (Bashkirova et al., 2018). The goal of this problem is to learn the bidirectional mappings between unpaired data in two different domains. Unlike the marginal distribution in each domain, learning a joint distribution is often ignored and has the following two critical challenges.

The first key challenge, from a probabilistic modeling perspective, is how to exploit the joint distribution of unpaired data by learning the bidirectional mappings between two different domains. In the unsupervised learning setting, there are two sets of samples drawn separately from two marginal distributions in two domains. Based on the coupling theory (Lindvall, 2002), there exist an infinite set of joint distributions given two marginal distributions, and hence infinite bidirectional mappings between two different domains. Therefore, directly learning the joint distribution without additional information between the marginal distributions is a highly ill-posed problem. Recently, many studies (Zhu et al., 2017; Yi et al., 2017; Kim et al., 2017) have been proposed to learn the mappings in two domains separately, which may incur the joint distribution mismatching issue. Therefore, how to exploit sufficient information from the joint distribution still remains an open question.

Another important challenge is that the generalization ability w.r.t. the learned joint distribution of two different domains is still unknown. Existing theoretical results (Pan et al., 2018; Galanti et al., 2018) ignore the joint distribution of different data and cannot guarantee the generalization ability of such joint distribution. Moreover, it is also very hard to evaluate the generalization ability practically. Regarding this issue, according to (Bojanowski et al., 2018), the generalization ability can be evaluated by the interpolation performance in the target domain. In this sense, we can extend image-to-image translation to video space by performing video interpolation in one domain and investigating the performance of the translated video in another domain. To achieve this, one may directly apply existing unsupervised image-to-image translation methods (Zhu et al., 2017; Kim et al., 2017; Yi et al., 2017). However, these methods may result in significantly incoherent videos with low visual quality. Therefore, it is important to design an effective joint distribution learning method and provide necessary theoretical analysis.

Regarding the above two challenges, in this paper, we propose a Joint Wasserstein Auto-Encoders (JWAE) to learn the bidirectional mappings between two domains by minimizing Wasserstein distance of joint distributions. Relying on the optimal transport theory, we are able to exploit sufficient information by matching latent distributions of images in two domains.

The contributions of this paper are summarized as follows:

- We propose a novel JWAE to solve the joint distribution matching problem. Based on Theorem 1, an intractable primal problem of optimal transport can be reduced to a simple optimization problem. More critically, our method is a generalization of CycleGAN (Liu et al., 2017) and UNIT (Liu et al., 2017).

- We provide a generalization bound of JWAE (see Theorem 4). In particular, we theoretically prove that the generalization ability of our method w.r.t. the learned joint distribution can be guaranteed by minimizing Wasserstein distance of joint distributions.

- To practically verify the generalization ability, we apply our method to unsupervised video-to-video synthesis and obtain two visually smooth videos in two different domains. Experiments on real-world datasets show the superiority of the proposed method over several state-of-the-arts.

## 2 RELATED WORK

In this paper, we consider the joint distribution matching problem. Recently, this problem has attracted extensive attention in image-to-image translation and video-to-video synthesis.

**Image-to-image translation.** Recently, Generative adversarial networks (GAN) (Goodfellow et al., 2014; Cao et al., 2018; Salimans et al., 2018), Variational Auto-Encoders (VAE) (Kingma & Welling, 2014) and Wasserstein Auto-Encoders (WAE) (Tolstikhin et al., 2017) have emerged as popular techniques for the image-to-image translation (I2IT) problem. For the unsupervised I2IT problem, CycleGAN (Zhu et al., 2017), DiscoGAN (Kim et al., 2017) and DualGAN (Yi et al., 2017) aim at minimizing the adversarial loss and the cycle-consistent loss in different domains, which may induce a joint distribution mismatching issue. To address this, CoGAN (Liu & Tuzel, 2016) learns a joint distribution by enforcing a weight-sharing constraint. Moreover, UNIT (Liu et al., 2017) builds upon CoGAN by using a shared-latent space assumption and the same weight-sharing constraint. However, these methods are not well-supported by any theoretical justifications.

**Video-to-video synthesis.** In this paper, we consider unsupervised video-to-video synthesis (V2VS) problem. Existing image-to-image methods (Zhu et al., 2017; Kim et al., 2017; Yi et al., 2017) cannot be directly used in the video-to-video synthesis problem, we further combine some video frame interpolation methods (Zhou et al., 2016; Ji et al., 2017; Niklaus et al., 2017; Liu et al., 2018) to synthesize video. Although UNIT (Liu et al., 2017) can be applied to video synthesis by interpolating in the latent space, it often results in temporally incoherent videos of low visual quality. Recently, a video-to-video translation method (Bashkirova et al., 2018) is proposed to translate a video in one domain to a video in another domain, but this method can not conduct video frame interpolation. Moreover, Wang et al. (2018) propose a video-to-video synthesis method and synthesize video results, but it cannot work for the unsupervised learning setting.

## 3 PROBLEM DEFINITION

**Notations**. We use calligraphic letters (*e.g.*, $\mathcal{X}$) for space, capital letters (*e.g.*, $X$) for random variables, and bold lower case letter (*e.g.*, $\mathbf{x}$) for their corresponding values. We denote probability distributions with capital letters (*i.e.*, $P(X)$) and corresponding densities with bold lower case letters (*i.e.*, $p(\mathbf{x})$). Let $(\mathcal{X}, P_X)$ be the domain, and $\mathcal{P}(\mathcal{X})$ be the set of all the probability measures over $\mathcal{X}$, and $P_X$ be the marginal distribution over $\mathcal{X}$. $\mathcal{S}_x = \{\mathbf{x}_i\}_{i=1}^M$ and $\mathcal{S}_y = \{\mathbf{y}_i\}_{i=1}^N$ are two sets of unpaired training data. For a set $\mathcal{S}$ and two functions $F: \mathcal{S} \rightarrow \mathbb{R}$ and $G: \mathcal{S} \rightarrow \mathbb{R}$, we denote $F(s) \lesssim G(s), \forall s \in \mathcal{S}$ if and only if $\exists \, C_1, C_2 > 0$ (independent of $s$) such that $F(s) \leq C_1 \cdot G(s) + C_2$.

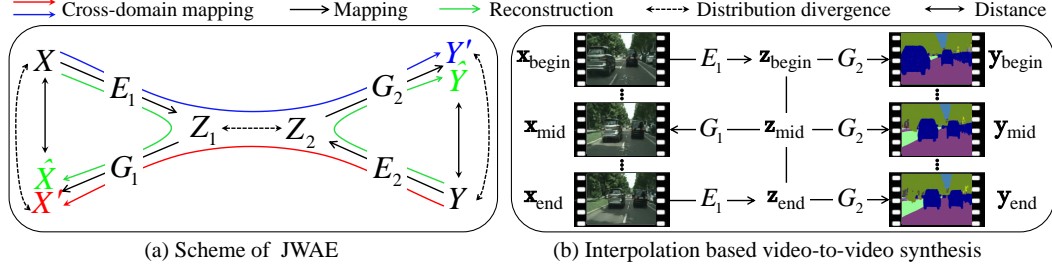

Figure 1: Demonstrations of (a) the JWAE scheme and (b) the interpolation based V2VS method.

### 3.1 JOINT DISTRIBUTION MATCHING PROBLEM

This paper considers the *Joint Distribution Matching Problem* for the unsupervised image-to-image translation task. Given two domains $(\mathcal{X}, P_X)$ and $(\mathcal{Y}, P_Y)$, where $P_X$ and $P_Y$ are distributions over $\mathcal{X}$ and $\mathcal{Y}$, our goal is to learn cross-domain mappings, *i.e.*, $f : \mathcal{X} \rightarrow \mathcal{Y}$ and $g : \mathcal{Y} \rightarrow \mathcal{X}$. Then we construct joint distributions in two domains, *i.e.*, $P_\mathcal{A}(X, f(X)), X \in \mathcal{X}$ and $P_\mathcal{B}(g(Y), Y), Y \in \mathcal{Y}$, and we minimize the following distribution divergence $d(P_\mathcal{A}, P_\mathcal{B})$ between these two joint distributions,

$$\min_{f \in \mathcal{F}, g \in \mathcal{G}} d(P_\mathcal{A}, P_\mathcal{B}), \tag{1}$$

where $\mathcal{F}$ and $\mathcal{G}$ are the sets of cross-domain mappings. Note that $d(\cdot, \cdot)$ is an arbitrary distribution divergence, *e.g.*, Wasserstein distance. In this paper, we study the joint distribution matching problem from the optimal transport (OT) point of view, which would bring some interesting results.

## 4 PROPOSED METHOD

### 4.1 WASSERSTEIN DISTANCE OF JOINT DISTRIBUTION

To address the joint distribution matching problem, one can learn a shared latent space $\mathcal{Z}$ for two different domains (Liu et al., 2017). In this sense, any pair of images in different domains can be mapped to the same latent representation. Inversely, there exist generative models $P_{G_1}(X'|Z)$ and $P_{G_2}(Y'|Z)$ that map a shared latent code $Z$ to $X'=G_1(Z)$ and $Y'=G_2(Z)$, respectively. Then $P_\mathcal{A}(X, Y')$ and $P_\mathcal{B}(X', Y)$ are two joint distributions between real and generated images, we can minimize Wasserstein distance $\mathcal{W}_c(P_\mathcal{A}, P_\mathcal{B})$ between joint distributions $P_\mathcal{A}$ and $P_\mathcal{B}$, *i.e.*,

$$\mathcal{W}_c(P_\mathcal{A}, P_\mathcal{B}) = \min_{P \in \mathcal{P}(P_\mathcal{A}, P_\mathcal{B})} \mathbb{E}_{(X, Y'; X', Y) \sim P}[c(X, Y'; X', Y)], \tag{2}$$

where $\mathcal{P}(P_\mathcal{A}, P_\mathcal{B})$ is the set of couplings which is composed of joint probability distributions with the probability distributions $(P_\mathcal{A}, P_\mathcal{B})$, and $c$ is any measurable cost function.

In practice, there are two important challenges on Wasserstein distance and the cost function. First, directly optimizing Problem (2) raises intractable computational difficulties (Genevay et al., 2018). Second, how to choose a appropriate cost function is very challenging. In this paper, we set $c(X, Y'; X', Y)=c_1(X, X')+c_2(Y', Y)$ (Bhushan Damodaran et al., 2018), where $c_1$ and $c_2$ can be any metric to measure the distance in $\mathcal{X} \times \mathcal{X}$ and $\mathcal{Y} \times \mathcal{Y}$, respectively. This helps to derive the following theorem so that the intractable Problem (2) can be reduced to a simple optimization problem.

**Theorem 1** *Given two deterministic models $P_{G_1}(X'|Z)$ and $P_{G_2}(Y'|Z)$ as Dirac measures, i.e., $P_{G_1}(X'|Z = \mathbf{z}) = \delta_{G_1(\mathbf{z})}$ and $P_{G_2}(Y'|Z = \mathbf{z}) = \delta_{G_2(\mathbf{z})}$ for all $\mathbf{z} \in \mathcal{Z}$, we have*

$$\mathcal{W}_c(P_\mathcal{A}, P_\mathcal{B}) = \inf_{Q \in \mathcal{Q}_1} \mathbb{E}_{P_X} \mathbb{E}_{Q(Z_1|X)}[c_1(X, G_1(Z_1))] + \inf_{Q \in \mathcal{Q}_2} \mathbb{E}_{P_Y} \mathbb{E}_{Q(Z_2|Y)}[c_2(G_2(Z_2), Y)], \tag{3}$$

*where $\mathcal{Q}_1=\{Q(Z_1|X)|Q_{Z_1}=P_Z=Q_{Z_2}, P_Y=P_{G_2}\}$ and $\mathcal{Q}_2=\{Q(Z_2|Y)|Q_{Z_1}=P_Z=Q_{Z_2}, P_X=P_{G_1}\}$ are the set of all probabilistic encoders, where $Q_{Z_1}$ and $Q_{Z_2}$ are the marginal distributions of $Z_1 \sim Q(Z_1|X)$ and $Z_2 \sim Q(Z_2|Y)$, where $X \sim P_X$ and $Y \sim P_Y$, respectively.* [1]

As previously mentioned, finding an optimal couplings between joint distributions $P_\mathcal{A}$ and $P_\mathcal{B}$ is very challenging. Fortunately, according to Theorem 1, we can instead optimize problem (3) for joint distribution matching. The details of objective functions and optimizations are given below.

---

[1] See supplementary materials for the proof.

## 4.2 Joint Wasserstein Auto-Encoders

As shown in Figure 1, we aim to learn the cross-domain mappings (*i.e.*, $G_2 \circ E_1$ and $G_1 \circ E_2$) from real data $X$ and $Y$ to samples $Y'$ and $X'$ such that the generated distributions are close to the real distribution, *i.e.*, $P_X = P_{G_1}$, $P_Y = P_{G_2}$. Moreover, the latent distributions generated by two Auto-Encoders (*i.e.*, $G_1 \circ E_1$ and $G_2 \circ E_2$) should be close to each other, *i.e.*, $Q_{Z_1} = Q_{Z_2}$. To this end, we optimize Problem (3) by relaxing the constraints $P_X = P_{G_1}$, $P_Y = P_{G_2}$ and $Q_{Z_1} = P_Z = Q_{Z_2}$. Then we minimize the regularized optimization problem:

$$\widehat{\mathcal{W}}_c(P_{\mathcal{A}}, P_{\mathcal{B}}) = \inf_Q \mathbb{E}_{P_X} \mathbb{E}_{Q(Z_1|X)}[c_1(X, G_1(Z_1))] + \inf_Q \mathbb{E}_{P_Y} \mathbb{E}_{Q(Z_2|Y)}[c_2(Y, G_2(Z_2))]$$
$$+ \alpha \mathcal{D}_X(P_X, P_{G_1}) + \beta \mathcal{D}_Y(P_Y, P_{G_2}) + \rho \mathcal{D}_Z(Q_{Z_1}, Q_{Z_2}), \quad (4)$$

where $\alpha, \beta, \rho$ are positive hyper-parameters, the last three terms can be arbitrary distribution divergences between two distributions. The above problems involves two kinds of functions, namely the reconstruction loss and distribution divergence.

**(i) Reconstruction loss.** We denote by $\mathcal{R}_x(E_1, E_2, G_1, G_2)$ and $\mathcal{R}_y(E_1, E_2, G_1, G_2)$ the empirical loss functions of the first two terms in Problem (4). Taking the case for the set $\mathcal{Q}_1$ as an example, the empirical reconstruction loss $\mathcal{R}_x$ can be rewritten as follows:

$$\mathcal{R}_x(E_1, E_2, G_1, G_2) = \frac{1}{M} \sum_{i=1}^{M} \underbrace{c_1(\mathbf{x}_i, G_1(E_1(\mathbf{x}_i)))}_{\text{Auto-Encoder loss}} + \underbrace{c_1(\mathbf{x}_i, G_1(E_2(G_2(E_1(\mathbf{x}_i)))))}_{\text{Cycle consistency loss}}, \quad (5)$$

Here, the first term represents the loss on the Auto-Encoders reconstruction and the second term inherently enforce the cycle consistency that widely studied in image-to-image translation tasks (Zhu et al., 2017; Liu et al., 2017; Kim et al., 2017; Yi et al., 2017). However, unlike existing methods, in our paper, the cycle consistency loss is directly derived from the joint distribution matching problem. The loss $\mathcal{R}_y$ can be constructed similarly. Last, let $\mathcal{R} = \mathcal{R}_x + \mathcal{R}_y$ be the final reconstruction loss.

**(ii) Distribution divergence.** The distribution divergences in Problem (4) can be measured by Adversarial loss (*e.g.*, original GAN, WGAN), Wasserstein distance, Maximum Mean Discrepancy (MMD) and Kullback-Leibler (KL) divergence, etc. Here, we use GAN to measure these divergences, denoted by $\text{GAN}(P_X, P_{G_1})$, $\text{GAN}(P_Y, P_{G_2})$ and $\text{GAN}(Q_{Z_1}, Q_{Z_2})$, respectively. Taking $\text{GAN}(P_X, P_{G_1})$ as example, the loss function $\mathcal{L}_x$ can be formulated as

$$\mathcal{L}_x(E_1, E_2, G_1, G_2, D_x) = \frac{1}{M} \sum_{i=1}^{M} 2\log(D_x(\mathbf{x}_i)) + \log(1 - D_x(G_1(E_2(G_2(E_1(\mathbf{x}_i))))))$$
$$+ \frac{1}{N} \sum_{i=1}^{N} \log(1 - D_x(G_1(E_2(\mathbf{y}_i)))). \quad (6)$$

Note that it is analogous to the form of Triple GAN (LI et al., 2017). We refer to the second term in the right-hand side of (6) as a cycle adversarial (CA) term [2]. Similarly, the losses $\mathcal{L}_y$ and $\mathcal{L}_z$ w.r.t. $\text{GAN}(P_Y, P_{G_2})$ and $\text{GAN}(Q_{Z_1}, Q_{Z_2})$ can be constructed. Please find the details in supplementary materials.

## 4.3 A concrete example: Interpolation based video-to-video synthesis

Since the generalization ability can be evaluated by the performance of interpolation (Bojanowski et al., 2018), we apply our method on the interpolation based video-to-video synthesis (V2VS) problem. The training and inference methods are shown in Algorithms 1 and 2, respectively.

In the training, given two sets of images $\{\mathbf{x}_i\}_{i=1}^{M}$ and $\{\mathbf{y}_j\}_{j=1}^{N}$ in two different domains, we seek to learn cross-domain mappings (see Algorithm 1). In the inference, given two input video frames $\mathbf{x}_{\text{begin}}$ and $\mathbf{x}_{\text{end}}$, we perform linear interpolation based video-to-video synthesis to produce two videos in two different domains. Specifically, we perform a linear interpolation between two embeddings extracted from the first domain and then decode it to a corresponding frame in the second domain (see Figure 1 (b) and Algorithm 2). In this sense, we can directly measure the quality of the synthesized video in the second domain to evaluate the generalization ability of the learned joint distribution mapping.

---

[2]Different from existing image-to-image translation methods, the cycle adversarial term is important for the performance of the proposed method. We will verify the effectiveness of this term in the experiments.

---

**Algorithm 1** Training details for JWAE.

---

**Input:** Training data in two different domains:
  $\{\mathbf{x}_i\}_{i=1}^M$ and $\{\mathbf{y}_j\}_{j=1}^N$.
**Initialization:** Models: $E_1, E_2, G_1, G_2$;
  Discriminators: $D_w, w \in \{x, y, z\}$.
  **repeat**
    Update $D_x, D_y, D_z$ by ascending:
      $\sum_w \mathcal{L}_w(E_1, E_2, G_1, G_2, D_w), \ w \in \{x, y, z\}$
    Update $E_1, E_2, G_1, G_2$ by descending:
      $\sum_w \mathcal{L}_w(E_1, E_2, G_1, G_2, D_w) + \mathcal{R}(E_1, E_2, G_1, G_2)$
  **until** models converged

---

**Algorithm 2** Inference for unsupervised V2VS.

---

**Input:** Testing data pair in the first domain:
  $\{\mathbf{x}_{\text{begin}}, \mathbf{x}_{\text{end}}\}$.
  Step 1: **Video frame interpolation**
    $\mathbf{z}_{\text{begin}} = E_1(\mathbf{x}_{\text{begin}}), \ \mathbf{z}_{\text{end}} = E_1(\mathbf{x}_{\text{end}})$
    $\mathbf{z}_{\text{mid}} = \alpha \mathbf{z}_{\text{begin}} + (1-\alpha)\mathbf{z}_{\text{end}}, \ \alpha \in (0,1)$
    $\mathbf{x}_{\text{mid}} = G_1(\mathbf{z}_{\text{mid}})$
    Synthesized video: $\{\mathbf{x}_{\text{begin}}, \mathbf{x}_{\text{mid}}, \mathbf{x}_{\text{end}}\}$
  Step 2: **Video translation**
    $\mathbf{y}_{\text{begin}} = G_2(\mathbf{z}_{\text{begin}}), \mathbf{y}_{\text{mid}} = G_2(\mathbf{z}_{\text{mid}}),$
    $\mathbf{y}_{\text{end}} = G_2(\mathbf{z}_{\text{end}})$
    Synthesized video: $\{\mathbf{y}_{\text{begin}}, \mathbf{y}_{\text{mid}}, \mathbf{y}_{\text{end}}\}$

---

## 5 GENERALIZATION ANALYSIS

In this section, we analyze generalization performance of JWAE. To begin with, we provide the definitions of the generalization error and probabilistic cross-domain Lipschitzness.

**Definition 1 (Generalization Error)** *Define cross-domain functions as $f = G_2 \circ E_1$ and $f = G_1 \circ E_2$ when $Q_{Z_1} = Q_{Z_2}$, and cost functions $c_1$ and $c_2$ which are bounded, symmetric, $L_c$-Lipschitz and satisfies the triangle inequality, and given two joint distributions $P_\mathcal{A}(X, Y)$ and $P_\mathcal{B}(X, Y)$, where $Y$ in $P_\mathcal{A}(X, Y)$ and $X$ in $P_\mathcal{B}(X, Y)$ are unknown, then the generalization error $E(f, g)$ becomes:*

$$E(f, g) = \mathbb{E}_{(X,Y) \sim P_\mathcal{A}(X,Y)} \left[c_2(Y, f(X))\right] + \mathbb{E}_{(X,Y) \sim P_\mathcal{B}(X,Y)} \left[c_1(X, g(Y))\right]. \tag{7}$$

Note that in image-to-image translation problem, it is common to assume that two close samples will have close outputs with high probability, *i.e.*, it satisfies a probabilistic Lipschitzness assumption (Courty et al., 2017; Urner et al., 2011). For convenience, we extend the definition as follows.

**Definition 2 ($\phi$-Probabilistic Cross-domain Lipschitzness)** *Given real and the generated marginal distribution $P_X$ and $P_{X'}$, and let $\phi : \mathbb{R}^+ \to [0, 1]$, we say that a function $f : \mathcal{X} \to \mathcal{Y}$ w.r.t. a joint distribution set $\mathcal{P}(P_X, P_{X'})$ over $P_X$ and $P_{X'}$ is $\phi$-Lipschitz if for all $\alpha > 0$,*

$$P_{(X,X') \sim \mathcal{P}(P_X, P_{X'})} \left[\|f(X) - f(X')\| < \alpha c_1(X, X')\right] \geq 1 - \phi(\alpha). \tag{8}$$

Intuitively, given a joint distribution set $\mathcal{P}(P_X, P_{X'})$, a function $f$ satisfying the $\alpha$-Lipschitz property holds with some probability. Then, we have the following results on the generalization error.

**Theorem 2** *Let $P^* = \arg\min_{P \in \mathcal{P}(P_\mathcal{A}^f, P_\mathcal{B}^g)} \mathbb{E}_{(X, Y^f; X^g, Y) \sim P}[c(X, Y^f; X^g, Y)]$ with Lipschitz cost functions $c_1$ and $c_2$. Let functions $f^* \in \mathcal{F}$ and $g^* \in \mathcal{G}$ be probabilistic cross-domain Lipschitzness w.r.t. $P^*$ that minimizes the joint error $E_{\mathcal{A},\mathcal{B}}(f^*, g^*)$. Given $M$ and $N$ instances drawn form $P_X$ and $P_Y$, respectively, with $L_{c_1}\alpha = L_{c_2}\beta = 1$, the following bound holds with probability at least $1 - \delta$:*

$$E(f, g) \lesssim \mathcal{W}(\widehat{P}_\mathcal{A}^f, \widehat{P}_\mathcal{B}^g) + \sqrt{\log\left(\frac{1}{\delta}\right)} \left(\frac{1}{\sqrt{M}} + \frac{1}{\sqrt{N}}\right) + E_{\mathcal{A},\mathcal{B}}(f^*, g^*) + \widetilde{\phi}(\alpha, \beta), \tag{9}$$

*where $E_{\mathcal{A},\mathcal{B}}(f^*, g^*) = E_\mathcal{A}(f^*) + E_\mathcal{B}^g(f^*) + E_\mathcal{B}(g^*) + E_\mathcal{A}^f(g^*)$, $E_\mathcal{A}(f^*) = \mathbb{E}_{(X,Y) \sim P_\mathcal{A}}[c_2(Y, f^*(X))]$, $E_\mathcal{A}^f(f^*) = \mathbb{E}_{(X,Y) \sim P_\mathcal{A}}[c_2(f(X), f^*(X))]$, and likewise for the definitions of $E_\mathcal{B}^g(f^*)$ and $E_\mathcal{B}(g^*)$. Here, $\widetilde{\phi}(\alpha, \beta) = L_{c_1}M_1\phi(\alpha) + L_{c_2}M_2\phi(\beta)$, where $\|f^*(X_1) - f^*(X_2)\| \leq M_1, \forall X_1, X_2 \in \mathcal{X}$ and $\|g^*(Y_1) - g^*(Y_2)\| \leq M_2, \forall Y_1, Y_2 \in \mathcal{Y}$.* [3]

**Remark 1** *Theorem 4 provides an upper bound on the generalization error. The first term in right hand of (9) corresponds to the empirical version of (2); While the second term means that we should minimize it with sufficient unpaired data from two different domains. The third term $E(f^*, g^*)$ correspond to the joint error. When the last term $\widetilde{\phi}(\alpha, \beta)$ is sufficiently small (i.e., $f$ and $g$ satisfy Lipschitz property with high probability), the cross-domain mappings would be well learned.*

---

[3]See supplementary materials for the detailed proof.

## 6 EXPERIMENTS

We apply our method to interpolation based video-to-video synthesis (V2VS) in the unsupervised setting. To be specific, we firstly conduct video interpolation between two input frames in one domain and then translate it to produce a corresponding video in another domain. The qualitative and quantitative results are shown in the following subsections [4].

**Datasets.** We conduct experiments on two widely used benchmark datasets, namely Cityscapes (Cordts et al., 2016) and SYNTHIA (Ros et al., 2016). (i) **Cityscapes** contains $2048 \times 1024$ street scene video of several German cities and a portion of ground truth semantic segmentation in the video. To obtain more semantic segmentation masks, following (Wang et al., 2018), we use a pre-trained DeepLab V3 network (Chen et al., 2017) to extract extra segmentation videos. (ii) We also study the generality of our algorithm on the unpaired dataset **SYNTHIA** (Ros et al., 2016), which contains a large collection of synthetic videos in different scenes and seasons. We perform unsupervised V2VS on four splits of SYNTHIA with different seasons, *i.e.*, spring, summer, fall and winter. In this paper, we adopt the winter split as the common domain and train models to translate videos from winter to the other three seasons.

**Evaluation metrics.** For quantitative comparisons, we adopt Fréchet Inception Distance (FID) (Heusel et al., 2017) to evaluate the quality of the frames in the synthesized videos. FID captures the similarity of the generated samples to real ones and correlates well with human judgement. Moreover, we also use a variant of FID (Wang et al., 2018) (FID4Video) to evaluate the quality of video. FID4Video measures the distribution similarity based on the extracted feature of videos. In general, for both FID and FID4Video, a lower score means the better performance.

### 6.1 IMPLEMENTATION DETAILS

We implement our method based on PyTorch[5]. We follow the experimental settings in Cycle-GAN (Zhu et al., 2017). For the optimization, we use Adam solver (Kingma & Ba, 2015) with a mini-batch size of 1 to train the models, and use a learning rate of 0.0002 for the first 100 epochs and gradually decrease it to zero for the next 100 epochs. Following (Zhu et al., 2017), we set $\alpha = \beta = 0.1$ in Eqn. (4). By default, we set $\rho = 0.1$ in our experiments.

### 6.2 BASELINE METHODS

We adopt several state-of-the-art baselines, including UNIT (Liu et al., 2017) with the latent space interpolation and several constructed variants of CycleGAN (Zhu et al., 2017) using different view synthesis algorithms. For those constructed baselines, we first conduct video interpolation and then perform image-to-image translation. Moreover, we also construct a variant of our method to conduct ablation study on the Triple GAN loss. All considered baselines are summarized as follows.

- **UNIT (Liu et al., 2017).** UNIT is a state-of-the-art unsupervised I2IT method which can perform supervised video-to-video synthesis by interpolating in the latent space.

- **DVF-Cycle.** This method combines the view synthesis method DVF (Liu et al., 2018) with CycleGAN (Zhu et al., 2017). To be specific, DVF produces videos by video interpolation in one domain. Then, we use CycleGAN to translate the generated video to another domain.

- **DVM-Cycle.** We use a geometrical view synthesis DVM (Ji et al., 2017) for video synthesis, and we replace DVF in DVF-Cycle with DVM and construct a new baseline called DVM-Cycle.

- **AdaConv-Cycle.** We also compare against a state-of-the-art video interpolation method Ada-Conv (Niklaus et al., 2017). For cross-domain video synthesis, we combine this method with a pre-trained CycleGAN model and term it AdaConv-CycleGAN in the following experiments.

- **W/O-CA.** To investigate the effect of the cycle adversarial (CA) terms in the loss functions (6) and (32), we construct a baseline method by removing it from the loss functions $\mathcal{L}_x$ and $\mathcal{L}_y$. We refer to this baseline as W/O-CA.

---

[4] Due to the page limit, more visual results are shown in the supplementary materials.
[5] PyTorch is from http://pytorch.org/.

Table 1: Performance comparisons with state-ot-the-art baselines on Cityscapes and SYNTHIA.

| Method | Cityscapes | | | | SYNTHIA | | | | | |
| | scene2segmentation | | segmentation2scene | | winter2spring | | winter2summer | | winter2fall | |
| | FID | FID4Video | FID | FID4Video | FID | FID4Video | FID | FID4Video | FID | FID4Video |
|---|---|---|---|---|---|---|---|---|---|---|
| DVF-Cycle | 110.59 | 23.95 | 151.27 | 40.61 | 152.44 | 42.22 | 160.69 | 42.43 | 163.13 | 41.04 |
| DVM-Cycle | 50.51 | 17.33 | 116.62 | 40.83 | 129.80 | 38.19 | 140.86 | 36.66 | 129.02 | 36.64 |
| AdaConv-Cycle | 33.50 | 14.96 | 99.67 | 30.24 | 117.40 | 23.83 | 126.01 | 20.62 | 110.52 | 16.77 |
| UNIT | 31.27 | 10.12 | 76.72 | 29.21 | 96.40 | 23.12 | 108.01 | 24.70 | 97.73 | 20.39 |
| W/O-CA | 24.32 | 8.34 | 47.41 | 27.37 | 92.77 | 21.83 | 84.83 | 19.54 | 91.37 | 15.87 |
| Ours | **22.74** | **6.80** | **43.48** | **25.87** | **88.24** | **21.37** | **77.12** | **17.99** | **87.50** | **14.14** |

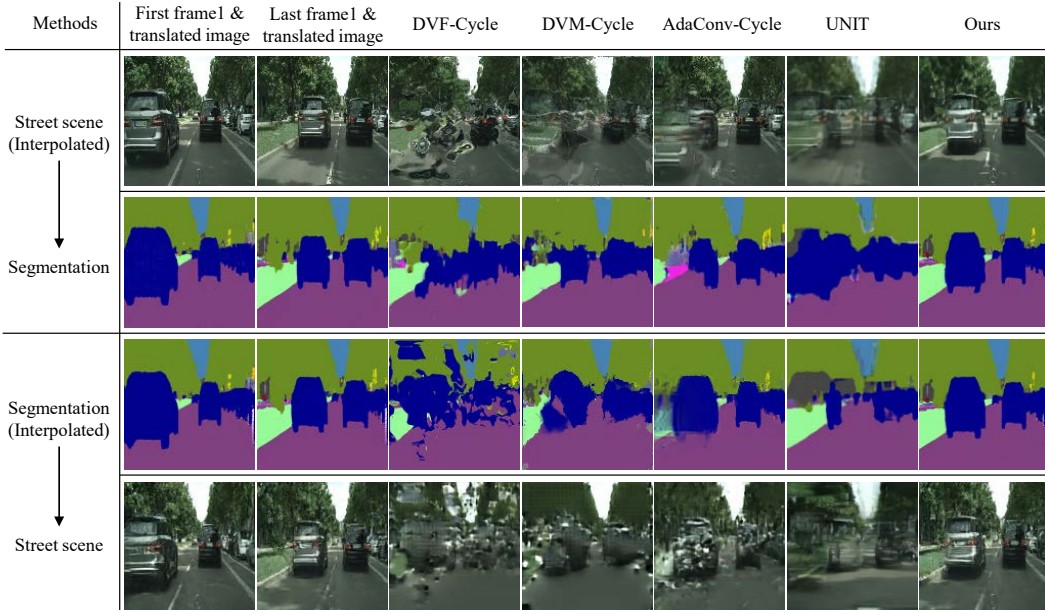

Figure 2: Comparisons of different methods for photo↔segmentation translation on Cityscapes dataset. We first synthesize a video of street scene and then translate it to the segmentation domain (Top), and vice versa for the mapping from segmentation to street scene (Bottom).

## 6.3 QUANTITATIVE COMPARISONS

We compare the performance on Cityscapes and SYNTHIA and show the results in Table 1. We can draw the following observations. **First**, our method consistently outperforms the baselines in terms of both FID and FID4Video scores. It means that our method produces frames and videos of promising quality and exhibits strong generalization ability. **Second**, with the help of Triple GAN loss, our method achieves better results than W/O-CA for both FID and FID4Video. This indicates that the reconstructed images after the cycle translation helps to learn a better joint distribution. The above observations demonstrate the superiority of our method over the competitive methods.

## 6.4 VISUAL COMPARISONS

**Visual results on Cityscapes.** We first interpolate videos in the cityscape domain and then translate them to the segmentation domain. In Figure 2, we compare the visual quality of both the interpolated and the translated images. From Figure 2 (top), our method produces sharper cityscape images and yields more accurate results in the semantic segmentation domain, which significantly outperforms the baseline methods, and vice versa in Figure 2 (bottom).

**Visual results on SYNTHIA.** We further evaluate the performance of our method on SYNTHIA. We synthesize videos among the domains of four seasons shown in Figure 3. **First**, our method is able to produce sharper images when interpolating the missing in-between frames (see top row of Figure 3). Second, the translated frames produced by our method in the other three seasons look

| Methods | First frame & translated image | Last frame & translated image | DVF-Cycle | DVM-Cycle | AdaConv-Cycle | UNIT | Ours |
|---|---|---|---|---|---|---|---|

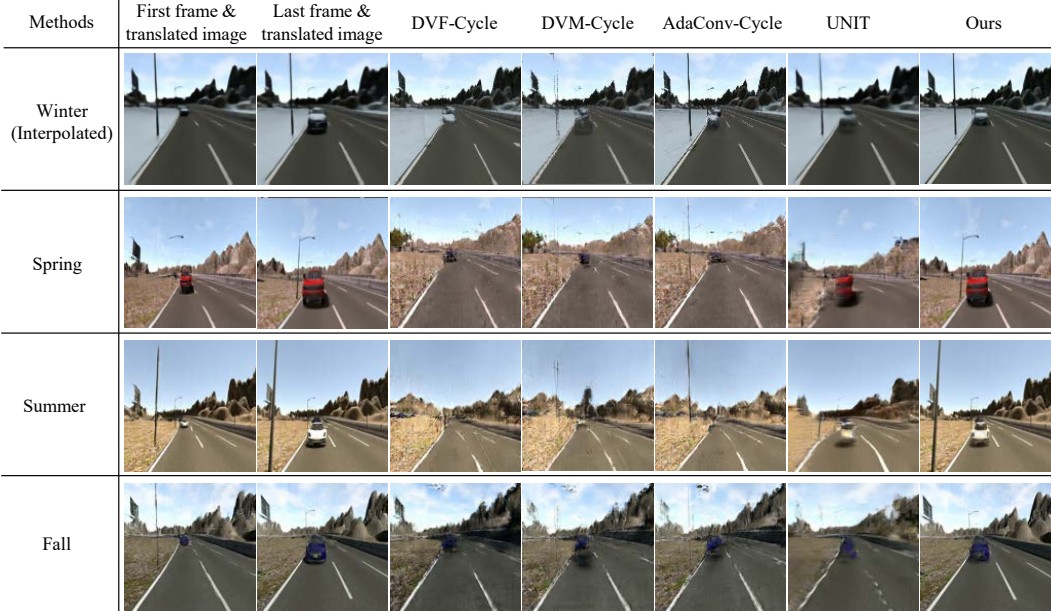

Figure 3: Comparison of different methods for season translation on SYNTHIA dataset. Top row: The synthesized video in the winter domain. Rows 2-4: The corresponding translated video in the domains of the other three seasons, *i.e.*, spring, summer and fall.

Table 2: Influence of $\rho$ for the adversarial loss on $\mathcal{Z}$ with different values. We compare the results of winter↔summer on SYNTHIA dataset in terms of FID and FID4Video scores.

| $\rho$ | winter2summer | | summer2winter | |
|---|---|---|---|---|
| | FID | FID4Video | FID | FID4Video |
| 0.01 | 94.91 | 20.29 | 107.65 | 18.90 |
| 0.1 | 77.12 | 17.99 | 89.03 | 17.36 |
| 1 | 89.07 | 21.04 | 102.18 | 18.63 |
| 10 | 101.07 | 23.66 | 108.47 | 20.50 |

more photo-realistic than those produced by the other baseline methods (see the shape of cars in rows 2-4 of Figure 3). These results demonstrate that our method is able to produce more promising videos and consistently outperforms other methods in different domains.

## 6.5 INFLUENCE OF $\rho$ FOR THE ADVERSARIAL LOSS ON $\mathcal{Z}$

We study the effect of the trade-off parameter $\rho$ over the adversarial loss on $\mathcal{Z}$ in Eqn. (4). The results are shown in Table 2. Given a very small weight $\rho$=0.01, the model obtains larger FID and FID4Video scores compared to that with $\rho$=0.1. When we increase it to $\rho = 1$ and $\rho = 10$, we also observe large performance degrades. Therefore, we suggest setting $\rho = 0.1$ in our method.

## 7 CONCLUSION

In this paper, we have proposed a novel joint Wasserstein Auto-Encoders method for the joint distribution matching problem. Instead of directly optimizing the primal problem of Wasserstein distance, we turn to propose a simple but effective optimization problem. In this way, we are able to conduct analysis on the generalization ability of JWAE and theoretically prove that minimizing the Wasserstein distance can guarantee the generalization ability. Extensive experiments on unsupervised V2VS task over several benchmark datasets demonstrate the superiority of the proposed method over the state-of-the-art methods.

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

# APPENDIX: "JOINT WASSERSTEIN AUTO-ENCODERS"

## A    PROOF OF THEOREM 1

**Proof**    We denote by $\mathcal{P}(X \sim P_X, X' \sim P_{G_1})$ and $\mathcal{P}(Y \sim P_Y, Y' \sim P_{G_2})$ the set of all joint distributions of $(X, X')$ and $(Y, Y')$ with marginals $P_X, P_{G_1}$ and $P_Y, P_{G_2}$, respectively, and denote by $\mathcal{P}(P_{\mathcal{A}}, P_{\mathcal{B}})$ the set of all joint distribution of $P_{\mathcal{A}}$ and $P_{\mathcal{B}}$. Recall the definition of Wasserstein distance $\mathcal{W}_c(P_{\mathcal{A}}, P_{\mathcal{B}})$, we have

$$\mathcal{W}_c(P_{\mathcal{A}}, P_{\mathcal{B}}) = \inf_{\pi \in \mathcal{P}(P_{\mathcal{A}}, P_{\mathcal{B}})} \mathbb{E}_{(X,Y';X',Y) \sim \pi}[c(X, Y'; X', Y)] \tag{10}$$

$$= \inf_{\pi \in \mathcal{P}(P_{\mathcal{A}}, P_{\mathcal{B}})} \mathbb{E}_{(X,Y';X',Y) \sim \pi}[c_1(X, X')] + \inf_{\pi \in \mathcal{P}(P_{\mathcal{A}}, P_{\mathcal{B}})} \mathbb{E}_{(X,Y';X',Y) \sim \pi}[c_2(Y', Y)]$$

$$= \inf_{P \in \mathcal{P}_{X,X'}} \mathbb{E}_{(X,X') \sim P}[c_1(X, X')] + \inf_{P \in \mathcal{P}_{Y,Y'}} \mathbb{E}_{(Y',Y) \sim P}[c_2(Y', Y)] \tag{11}$$

$$= \inf_{P \in \mathcal{P}(P_X, P_{G_1})} \mathbb{E}_{(X,X') \sim P}[c_1(X, X')] + \inf_{P \in \mathcal{P}(P_Y, P_{G_2})} \mathbb{E}_{(Y',Y) \sim P}[c_2(Y', Y)] \tag{12}$$

$$= \mathcal{W}_{c_1}(P_X, P_{G_1}) + \mathcal{W}_{c_2}(P_{G_2}, P_Y).$$

Line (10) holds by the definition of $\mathcal{W}_c(P_{\mathcal{A}}, P_{\mathcal{B}})$. Line (11) uses the fact that the variable pair $(X, X')$ is independent of the variable pair $(Y, Y')$, and $\mathcal{P}_{X,X'}$ and $\mathcal{P}_{Y,Y'}$ are the marginals on $(X, X')$ and $Y, Y'$ induced by joint distributions in $\mathcal{P}_{X,Y',X',Y}$. In Line (12), if $P_{G_1}(X'|Z)$ and $P_{G_2}(Y'|Z)$ are Dirac measures (*i.e.*, $X' = G_1(Z)$ and $Y' = G_2(Z)$), we have

$$\mathcal{P}_{X,X'} = \mathcal{P}(P_X, P_{G_1}), \qquad \mathcal{P}_{Y,Y'} = \mathcal{P}(P_Y, P_{G_2}).$$

We consider certain sets of joint probability distributions $\mathcal{P}_{X,X',Z_1}$ and $\mathcal{P}_{Y,Y',Z_2}$ of three random variables $(X, X', Z_1) \in \mathcal{X} \times \mathcal{X} \times \mathcal{Z}$ and $(Y', Y, Z_2) \in \mathcal{Y} \times \mathcal{Y} \times \mathcal{Z}$, respectively. We denote by $\mathcal{P}(X \sim P_X, Z_1 \sim P_{Z_1})$ and $\mathcal{P}(Y \sim P_Y, Z_2 \sim P_{Z_2})$ the set of all joint distributions of $(X, Z_1)$ and $(Y, Z_2)$ with marginals $P_X, P_{Z_1}$ and $P_Y, P_{Z_2}$, respectively. The set of all joint distributions $\mathcal{P}_{X,X',Z_1}$ such that $X \sim P_X$, $(X', Z_1) \sim P_{G_1,Z_1}$ and $(X' \perp\!\!\!\perp X)|Z_1$, and likewise for $\mathcal{P}_{Y,Y',Z_2}$. We denote by $\mathcal{P}_{X,X'}$ and $\mathcal{P}_{X,Z_1}$ the sets of marginals on $(X, X')$ and $(X, Z_1)$ induced by distributions in $\mathcal{P}_{X,X',Z_1}$, respectively, and likewise for $\mathcal{P}_{Y,Y'}$ and $\mathcal{P}_{Y,Z_2}$. For the further analyses, we have

$$\mathcal{W}_{c_1}(P_X, P_{G_1}) + \mathcal{W}_{c_2}(P_Y, P_{G_2})$$

$$= \inf_{P \in \mathcal{P}_{X,X',Z_1}} \mathbb{E}_{(X,X',Z_1) \sim P}[c_1(X, X')] + \inf_{P \in \mathcal{P}_{Y',Y,Z_2}} \mathbb{E}_{(Y',Y,Z_2) \sim P}[c_2(Y', Y)] \tag{13}$$

$$= \inf_{P \in \mathcal{P}_{X,X',Z_1}} \mathbb{E}_{P_{Z_1}} \mathbb{E}_{X \sim P(X|Z_1)} \mathbb{E}_{X' \sim P(X'|Z_1)}[c_1(X, X')] \tag{14}$$

$$+ \inf_{P \in \mathcal{P}_{Y',Y,Z_2}} \mathbb{E}_{P_{Z_2}} \mathbb{E}_{Y \sim P(Y|Z_2)} \mathbb{E}_{Y' \sim P(Y'|Z_2)}[c_2(Y', Y)]$$

$$= \inf_{P \in \mathcal{P}_{X,X',Z_1}} \mathbb{E}_{P_{Z_1}} \mathbb{E}_{X \sim P(X|Z_1)}[c_1(X, G_1(Z_1))] + \inf_{P \in \mathcal{P}_{Y',Y,Z_2}} \mathbb{E}_{P_{Z_2}} \mathbb{E}_{Y \sim P(Y|Z_2)}[c_2(G_2(Z_2), Y)]$$

$$= \inf_{P \in \mathcal{P}_{X,Z_1}} \mathbb{E}_{(X,Z_1) \sim P}[c_1(X, G_1(Z_1))] + \inf_{P \in \mathcal{P}_{Y,Z_2}} \mathbb{E}_{(Y,Z_2) \sim P}[c_2(G_2(Z_2), Y)] \tag{15}$$

$$= \inf_{P \in \mathcal{P}(X,Z_1)} \mathbb{E}_{(X,Z_1) \sim P}[c_1(X, G_1(Z_1))] + \inf_{P \in \mathcal{P}(Y,Z_2)} \mathbb{E}_{(Y,Z_2) \sim P}[c_2(G_2(Z_2), Y)] \tag{16}$$

$$= \inf_{Q \in \mathcal{Q}_1} \mathbb{E}_{P_X} \mathbb{E}_{Q(Z_1|X)}[c_1(X, G_1(Z_1))] + \inf_{Q \in \mathcal{Q}_2} \mathbb{E}_{P_Y} \mathbb{E}_{Q(Z_2|Y)}[c_2(G_2(Z_2), Y)], \tag{17}$$

where $\mathcal{Q}_1 = \{Q(Z_1|X)|Q_{Z_1} = P_Z = Q_{Z_2}, P_Y = P_{G_2}\}$ and $\mathcal{Q}_2 = \{Q(Z_2|Y)|Q_{Z_1} = P_Z = Q_{Z_2}, P_X = P_{G_1}\}$ are the set of all probabilistic encoders, where $Q_{Z_1}$ and $Q_{Z_2}$ are the marginal distributions of $Z_1 \sim Q(Z_1|X)$ and $Z_2 \sim Q(Z_2|Y)$, where $X \sim P_X$ and $Y \sim P_Y$, respectively.

Line (13) uses the tower rule of expectation and Line (14) holds by the conditional independence property of $\mathcal{P}_{X,X',Z}$. In line (15), we take the expectation w.r.t. $X'$ and $Y'$, respectively, and use the total probability. Line (16) follows the fact that $\mathcal{P}_{X,Z_1} = \mathcal{P}(X \sim P_X, Z_1 \sim P_{Z_1})$ and $\mathcal{P}_{Y,Z_2} = \mathcal{P}(Y \sim P_Y, Z_2 \sim P_{Z_2})$ since $\mathcal{P}(P_X, P_{G_1}), \mathcal{P}_{X,X',Z_1}$ and $\mathcal{P}_{X,Y}$ depend on the choice of conditional distributions $P_{G_1}(X'|Z_1)$, while $P_{X,Z_1}$ does not, and likewise for distributions w.r.t. $Y$ and $G_2$. In line (17), the generative model $Q(Z_1|X)$ can be derived from two cases where $Z_1$ can be sampled from $E_1(X)$ and $E_2(G_2(E_1(X)))$ when $Q_{Z_1} = Q_{Z_2}$ and $P_Y = P_{G_2}$, and likewise for the generative model $Q(Z_2|Y)$.    $\square$

## B  PROOF OF THEOREM 4

**Lemma 1** *Given an $L_{c_2}$-Lipschitz cost function $c_2$, a probabilistic cross-domain Lipschitz function $f^*$, and two joint distributions $P_{\mathcal{A}}^f$ and $P_{\mathcal{B}}^g$ w.r.t. $f$ and $g$, respectively, then for any coupling $P(X, Y^f; X^g, Y) \in \mathcal{P}(P_{\mathcal{A}}^f, P_{\mathcal{B}}^g)$, we have*

$$\left| \int_{\mathcal{X} \times \mathcal{Y}} c_2(Y, f^*(X)) d(P_{\mathcal{A}}^f - P_{\mathcal{B}}^g) \right| \leq \int_{(\mathcal{X} \times \mathcal{Y})^2} \left| c_2(Y^f, f^*(X)) - c_2(Y, f^*(X^g)) \right| dP(X, Y^f; X^g, Y),$$

*where $P_{\mathcal{A}}^f = (X, Y^f)_{X \sim P_X}$ and $P_{\mathcal{B}}^g = (X^g, Y)_{Y \sim P_Y}$, and $Y^f = f(X)$ and $X^g = g(Y)$.*

**Proof**  Given joint distributions $P_{\mathcal{A}}^f = (X, f(X))_{X \sim P_X}$ and $P_{\mathcal{B}}^g = (g(Y), Y)_{Y \sim P_Y}$, and an $L_{c_2}$-Lipschitz cost function $c_2$ on $\mathcal{Y} \times \mathcal{Y}$, and based on the duality form of the Kantorovitch-Rubinstein theorem (Rachev et al., 1990; Villani, 2008), we have

$$\sup \int_{\mathcal{X} \times \mathcal{Y}} c_2(Y, f^*(X)) d(P_{\mathcal{A}}^f - P_{\mathcal{B}}^g) = \inf \int_{(\mathcal{X} \times \mathcal{Y})^2} d_{f^*}(X, Y^f; X^g, Y) dP(X, Y^f; X^g, Y)$$

where $d_{f^*}(X, Y^f; X^g, Y)$ is a cost function of $f^*$ w.r.t. $(X, Y^f; X^g, Y)$, and the cost function $c_2$ satisfies the condition, *i.e.*, $c_2(Y^f, f^*(X)) - c_2(Y, f^*(X^g)) \leq d_{f^*}(X, Y^f; X^g, Y)$. For simplicity, we choose its equality, then we have

$$\left| \int_{\mathcal{X} \times \mathcal{Y}} c_2(Y, f^*(X)) d(P_{\mathcal{A}}^f - P_{\mathcal{B}}^g) \right|$$

$$\leq \left| \int_{(\mathcal{X} \times \mathcal{Y})^2} c_2(Y^f, f^*(X)) - c_2(Y, f^*(X^g)) dP(X, Y^f; X^g, Y) \right|$$

$$\leq \int_{(\mathcal{X} \times \mathcal{Y})^2} \left| c_2(Y^f, f^*(X)) - c_2(Y, f^*(X^g)) \right| dP(X, Y^f; X^g, Y).$$

$\square$

**Lemma 2** *Given an $L_{c_1}$-Lipschitz cost function $c_1$, two joint distributions $P_{\mathcal{A}}^f$ and $P_{\mathcal{B}}^g$, and a $\phi(\alpha)$-probabilistic cross-domain Lipschitz function $f^*$, we assume the input space $\mathcal{X}$ is bounded such that $\|f^*(X_1) - f^*(X_2)\| \leq M_1$, then we have*

$$\left| E_{\mathcal{A}}^f(f^*) - E_{\mathcal{B}}^g(f^*) \right| \leq \mathcal{W}(P_{\mathcal{A}}^f, P_{\mathcal{B}}^g) + L_{c_1} M_1 \phi(\alpha).$$

**Proof**  Based on the definition of $E_{\mathcal{A}}^f(f^*)$ and $E_{\mathcal{B}}^g(f^*)$, we discuss the absolute value of the difference between them as follows:

$$\left| E_{\mathcal{A}}^f(f^*) - E_{\mathcal{B}}^g(f^*) \right|$$

$$= \left| \mathbb{E}_{(X,Y) \sim P_{\mathcal{A}}^f(X,Y)} [c_2(Y, f^*(X))] - \mathbb{E}_{(X,Y) \sim P_{\mathcal{B}}^g(X,Y)} [c_2(Y, f^*(X))] \right|$$

$$= \left| \int_{\mathcal{X} \times \mathcal{Y}} c_2(Y, f^*(X)) d\left( P_{\mathcal{A}}^f - P_{\mathcal{B}}^g \right) \right| \tag{18}$$

$$\leq \int_{(\mathcal{X} \times \mathcal{Y})^2} \left| c_2(Y^f, f^*(X)) - c_2(Y, f^*(X^g)) \right| dP^*(X, Y^f; X^g, Y) \tag{19}$$

$$= \int_{(\mathcal{X} \times \mathcal{Y})^2} \left| c_2(Y^f, f^*(X)) - c_2(Y^f, f^*(X^g)) + c_2(Y^f, f^*(X^g)) - c_2(Y, f^*(X^g)) \right| dP^*(X, Y^f; X^g, Y)$$

$$\leq \int_{(\mathcal{X} \times \mathcal{Y})^2} \left| c_2(Y^f, f^*(X)) - c_2(Y^f, f^*(X^g)) \right| + \left| c_2(Y^f, f^*(X^g)) - c_2(Y, f^*(X^g)) \right| dP^*(X, Y^f; X^g, Y)$$

$$\leq \int_{(\mathcal{X} \times \mathcal{Y})^2} \left[ L_{c_2} \| f^*(X) - f^*(X^g) \| + c_2(Y^f, Y) \right] dP^*(X, Y^f; X^g, Y) \tag{20}$$

$$\leq \int_{(\mathcal{X} \times \mathcal{Y})^2} \left[ L_1 \alpha c_1(X, X^g) + c_2(Y^f, Y) \right] dP^*(X, Y^f; X^g, Y) + L_1 M_1 \phi(\alpha) \tag{21}$$

$$\leq \int_{(\mathcal{X} \times \mathcal{Y})^2} \left[ c_1(X^g, X) + c_2(Y, Y^f) \right] dP^*(X, Y^f; X^g, Y) + L_1 M_1 \phi(\alpha) \tag{22}$$

$$= \mathcal{W}(P_{\mathcal{A}}^f, P_{\mathcal{B}}^g) + L_1 M_1 \phi(\alpha). \tag{23}$$

Line (18) follows by the definition of $E_{\mathcal{A}}^f(f^*)$ and $E_{\mathcal{B}}^g(f^*)$ and the definition of the expectation w.r.t. $P_{\mathcal{A}}^f$ and $P_{\mathcal{B}}^g$. Line (19) holds by Lemma 1 and we choose the optimal coupling $P^*$, *i.e.*,

$$P^* = \arg\min_{P \in \mathcal{P}(P_{\mathcal{A}}^f, P_{\mathcal{B}}^g)} \mathbb{E}_{(X,Y';X',Y) \sim P}[c(X,Y';X',Y)].$$

Line (20) holds by the $L_{c_2}$-Lipschitz cost function $c_2$ w.r.t. the second argument, *i.e.*,

$$\left| c_2(Y^f, f^*(X)) - c_2(Y^f, f^*(X^g)) \right| \le L_{c_2} \left| f^*(X) - f^*(X^g) \right|,$$

and the triangle inequality of the cost $c_2$, *i.e.*, $|c_2(Y^f, f^*(X^g)) - c_2(Y, f^*(X^g))| \le c_2(Y^f, Y)$. Based on Definition 2, line (21) contains two cases: (1) $f^*$ and $\mathcal{P}^*$ satisfies the $\phi$-probabilistic cross-domain Lipschitzness with probability $1 - \phi(\alpha)$; (2) the difference of $f^*$ between any two instances is bounded by $M_1$, and we have the additional term $L_{c_1} M_1 \phi(\alpha)$ that covers the regions where the $\phi$-probabilistic cross-task Lipschitzness does not hold, *i.e.*

$$\int_{(\mathcal{X} \times \mathcal{Y})^2} L_{c_1} |f^*(X) - f^*(X^g)| \, dP^*(X, Y^f; X^g, Y)$$
$$\le \int_{(\mathcal{X} \times \mathcal{Y})^2} L_{c_1} \alpha c_2(X, X^g) dP^*(X, Y^f; X^g, Y) + L_{c_1} M_1 \phi(\alpha).$$

Line (22) holds by the Lipschitz cost function $c_1$, and line (23) uses the definition of $\mathcal{W}(P_{\mathcal{A}}^f, P_{\mathcal{B}}^g)$. $\square$

**Lemma 3** *Given Lipschitz cost function $c_1$ and $c_2$, and the input instances are bounded for $f^*$ and $g^*$, i.e., $\|f^*(\mathbf{x}_1) - f^*(\mathbf{x}_2)\| \le M_1$ and $\|g^*(\mathbf{y}_1) - g^*(\mathbf{y}_2)\| \le M_2$, if $L_{c_1}\alpha = L_{c_2}\beta = 1$, we have*

$$E_{\mathcal{A}}(f) \le \mathcal{W}_c(P_{\mathcal{A}}^f, P_{\mathcal{B}}^g) + L_{c_1} M_1 \phi(\alpha) + E_{\mathcal{A}}(f^*) + E_{\mathcal{B}}^g(f^*),$$
$$E_{\mathcal{B}}(g) \le \mathcal{W}_c(P_{\mathcal{A}}^f, P_{\mathcal{B}}^g) + L_{c_2} M_2 \phi(\beta) + E_{\mathcal{B}}(g^*) + E_{\mathcal{A}}^f(g^*),$$

*with probability at least $1 - \delta$.*

**Proof**   Given $E_{\mathcal{A}}(f) = \mathbb{E}_{(X,Y) \sim P_{\mathcal{A}}}[c_2(Y, f(X))]$ and $E_{\mathcal{B}}(g) = \mathbb{E}_{(X,Y) \sim P_{\mathcal{B}}}[c_1(X, g(Y))]$, without loss of generality, we analyze the former as follows, and the latter is similar for the results.

$$
\begin{aligned}
E_{\mathcal{A}}(f) =& \mathbb{E}_{(X,Y) \sim P_{\mathcal{A}}}[c_2(Y, f(X))] & \\
\le& \mathbb{E}_{(X,Y) \sim P_{\mathcal{A}}}[c_2(Y, f^*(X)) + c_2(f^*(X), f(X))] & (24) \\
=& \mathbb{E}_{(X,Y) \sim P_{\mathcal{A}}}[c_2(f(X), f^*(X))] + E_{\mathcal{A}}(f^*) & (25) \\
=& \mathbb{E}_{(X,Y) \sim P_{\mathcal{A}}^f}[c_2(f(X), f^*(X)] + E_{\mathcal{A}}(f^*) & (26) \\
=& E_{\mathcal{A}}^f(f^*) + E_{\mathcal{A}}(f^*) & \\
=& E_{\mathcal{A}}^f(f^*) - E_{\mathcal{B}}^g(f^*) + E_{\mathcal{B}}^g(f^*) + E_{\mathcal{A}}(f^*) & \\
\le& \left| E_{\mathcal{A}}^f(f^*) - E_{\mathcal{B}}^g(f^*) \right| + E_{\mathcal{B}}^g(f^*) + E_{\mathcal{A}}(f^*) & (27) \\
\le& \mathcal{W}_c(P_{\mathcal{A}}^f, P_{\mathcal{B}}^g) + L_1 M_1 \phi(\alpha) + E_{\mathcal{B}}^g(f^*) + E_{\mathcal{A}}(f^*). & (28)
\end{aligned}
$$

Line (24) holds by an assumption that the loss function $c_1$ satisfies triangle inequality. Line (25) holds by a symmetric loss function $c_1$ and the definition of $E_{\mathcal{A}}(f^*)$, *i.e.*, $E_{\mathcal{A}}(f^*) = \mathbb{E}_{(X,Y) \sim P_{\mathcal{A}}}[c_2(Y, f^*(X)]$. Line (26) uses a fact that

$$\mathbb{E}_{(X,Y) \sim P_{\mathcal{A}}}[c_2(f(X), f^*(X))] = \mathbb{E}_{(X, f(X)) \sim P_{\mathcal{A}}^f}[c_2(f(X), f^*(X))] = E_{\mathcal{A}}^f(f^*).$$

$\square$

In order to prove Theorem 4, we introduce an important theorem that shows the concentration inequality for Wasserstein distance of the empirical measure and its true measure.

**Theorem 3 (Empirical Concentration (Bolley et al., 2007), Theorem 1.1)** *Let $\mu$ be a probability measure on $\Omega$ and $\hat{\mu} = \frac{1}{N} \sum_{i=1}^{N} \delta_{\mathbf{w}_i}$ be the associated empirical measure defined on a sample of independent variables $\{\mathbf{w}_i\}_{i=1}^{N}$ drawn from $\mu$. Then, for any $d' > dim(\Omega)$ and $C' < C$, there exists*

some constant $N_0$ depending on $d'$ and some square exponential monments of $\mu$ such that for any $\epsilon > 0$ and $N \geq N_0 \max\{\epsilon^{-(d'+2)}, 1\}$,

$$P\left[\mathcal{W}(\mu, \widehat{\mu}) > \epsilon\right] \leq e^{-\frac{C'}{2}N\epsilon^2},$$

where $C'$ can be calculated explicitly, and $C$ is such that $\mu$ verifies for any measure $\nu$ the Talagrand (transportation) inequality $T_1(C) : \mathcal{W}(\mu, \nu) \leq (\frac{2}{C}H(\nu|\mu))^{\frac{1}{2}}$ with the relative entropy $H$, and $T_1(C)$ holds when for some $\tau > 0$ and $\int e^{\tau d(\mathbf{w}, \mathbf{w}')^2} d\mu(\mathbf{w}) < +\infty, \forall \mathbf{w}' \in \Omega$.

This theorem allows us to propose generalization bounds based on the Wasserstein distance. Now we can use this theorem and the previous Lemma to prove the following theorem.

**Theorem 4** Let $P^* = \arg\min_{P \in \mathcal{P}(P_\mathcal{A}^f, P_\mathcal{B}^g)} \mathbb{E}_{(X, Y^f; X^g, Y) \sim P}[c(X, Y^f; X^g, Y)]$ with Lipschitz cost functions $c_1$ and $c_2$. Let functions $f^* \in \mathcal{F}$ and $g^* \in \mathcal{G}$ be probabilistic cross-domain Lipschitzness w.r.t. $P^*$ that minimizes the joint error $E_{\mathcal{A}, \mathcal{B}}(f^*, g^*)$. Given $M$ and $N$ instances drawn form $P_X$ and $P_Y$, respectively, with $L_{c_1}\alpha = L_{c_2}\beta = 1$, the following bound holds with probability at least $1 - \delta$:

$$E(f, g) \leq 2W_1(\widehat{P}_\mathcal{A}^f, \widehat{P}_\mathcal{B}^g) + \sqrt{\frac{2}{C'}\log\left(\frac{2}{\delta}\right)}\left(\frac{2}{\sqrt{M}} + \frac{2}{\sqrt{N}}\right) + E_{\mathcal{A}, \mathcal{B}}(f^*, g^*) + \tilde{\phi}(\alpha, \beta),$$

where $E_{\mathcal{A}, \mathcal{B}}(f^*, g^*) = E_\mathcal{A}(f^*) + E_\mathcal{B}^g(f^*) + E_\mathcal{B}(g^*) + E_\mathcal{A}^f(g^*)$, $E_\mathcal{A}(f^*) = \mathbb{E}_{(X,Y) \sim P_\mathcal{A}}[c_2(Y, f^*(X))]$, $E_\mathcal{A}^f(f^*) = \mathbb{E}_{(X,Y) \sim P_\mathcal{A}}[c_2(f(X), f^*(X))]$, and likewise for the definitions of $E_\mathcal{B}^g(f^*)$ and $E_\mathcal{B}(g^*)$. Here, $\tilde{\phi}(\alpha, \beta) = L_{c_1}M_1\phi(\alpha) + L_{c_2}M_2\phi(\beta)$, where $\|f^*(X_1) - f^*(X_2)\| \leq M_1, \forall X_1, X_2 \in \mathcal{X}$ and $\|g^*(Y_1) - g^*(Y_2)\| \leq M_2, \forall Y_1, Y_2 \in \mathcal{Y}$. Here, $C'$ is some constant satisfying Theorem 3.

**Proof** Based on the definition of $E(f, g)$, we have

$$E(f, g) = \mathbb{E}_{(X,Y) \sim P_\mathcal{A}}[c_2(Y, f(X))] + \mathbb{E}_{(X,Y) \sim P_\mathcal{B}}[c_1(X, g(Y))]$$

$$\leq 2\mathcal{W}(P_\mathcal{A}^f, P_\mathcal{B}^g) + E(f^*, g^*) + \tilde{\phi}(\alpha, \beta) \tag{29}$$

$$\leq 2\mathcal{W}(\widehat{P}_\mathcal{A}^f, \widehat{P}_\mathcal{B}^g) + \mathcal{W}(\widehat{P}_\mathcal{A}^f, P_\mathcal{A}^f) + \mathcal{W}(\widehat{P}_\mathcal{B}^g, P_\mathcal{B}^g) + E(f^*, g^*) + \tilde{\phi}(\alpha, \beta) \tag{30}$$

$$\leq 2\mathcal{W}(\widehat{P}_\mathcal{A}^f, \widehat{P}_\mathcal{B}^g) + \sqrt{\frac{2}{t'}\log\left(\frac{2}{\delta}\right)}\left(\frac{2}{\sqrt{M}} + \frac{2}{\sqrt{N}}\right) + E(f^*, g^*) + \tilde{\phi}(\alpha, \beta). \tag{31}$$

Line (29) holds by the conclusion in Lemma 3, and line (30) uses the triangle inequality for 1-Wasserstein distance. In line (31), we apply the union bound of $\mathcal{W}(\widehat{P}_\mathcal{A}^f, P_\mathcal{A}^f)$ and $\mathcal{W}(\widehat{P}_\mathcal{B}^g, P_\mathcal{B}^g)$ (with probability $\frac{\delta}{2}$ for each) in Theorem 3. $\square$

## C DETAILS OF OPTIMIZATION FOR $\text{GAN}_Y(P_Y, P_{G_2})$ AND $\text{GAN}_Z(Q_{Z_1}, Q_{Z_2})$

In this section, we discuss some details of $\text{GAN}(P_Y, P_{G_2})$ and $\text{GAN}(Q_{Z_1}, Q_{Z_2})$ as follows:

(1) For $\text{GAN}(P_Y, P_{G_2})$, the loss function $\mathcal{L}_y$ can be written as the following optimization problem:

$$\mathcal{L}_y(E_1, E_2, G_1, G_2, D_y) = \frac{1}{N}\sum_{i=1}^{N} 2\log(D_y(\mathbf{y}_i)) + \log(1 - D_y(G_2(E_1(G_1(E_2(\mathbf{y}_i)))))) \\ + \frac{1}{M}\sum_{i=1}^{M}\log(1 - D_y(G_2(E_1(\mathbf{x}_i)))). \tag{32}$$

(2) For $\text{GAN}(Q_{Z_1}, Q_{Z_2})$, we decompose it into $\text{GAN}_Z(P_Z, Q_{Z_1})$ and $\text{GAN}_Z(P_Z, Q_{Z_2})$, denoted by $\mathcal{L}_{z_1}$ and $\mathcal{L}_{z_2}$. Then we simultaneously optimize these two loss functions $\mathcal{L}_{z_1}$ and $\mathcal{L}_{z_2}$, and they can be formulated as the following optimization problems:

$$\mathcal{L}_{z_1}(E_1, E_2, G_2, D_z) = \frac{1}{M}\sum_{i=1}^{M} 2\log(D_z(\mathbf{z}_i)) + \log(1 - D_z(E_1(\mathbf{x}_i))), \tag{33}$$

$$\mathcal{L}_{z_2}(E_1, E_2, G_1, D_z) = \frac{1}{N}\sum_{i=1}^{N} 2\log(D_z(\mathbf{z}_i)) + \log(1 - D_z(E_2(\mathbf{y}_i))), \tag{34}$$

where $\mathbf{z}$ is drawn from the prior distribution $P_Z$, e.g., Gaussian distribution. Last, we let

$$\mathcal{L}_z(E_1, E_2, G_1, G_2, D_z) = \mathcal{L}_{z_1}(E_1, E_2, G_2, D_z) + \mathcal{L}_{z_2}(E_1, E_2, G_1, D_z)$$

be the final loss for $\text{GAN}(Q_{Z_1}, Q_{Z_2})$.

## D  VISUAL RESULTS ON SYNTHIA AND CITYSCAPES DATASETS

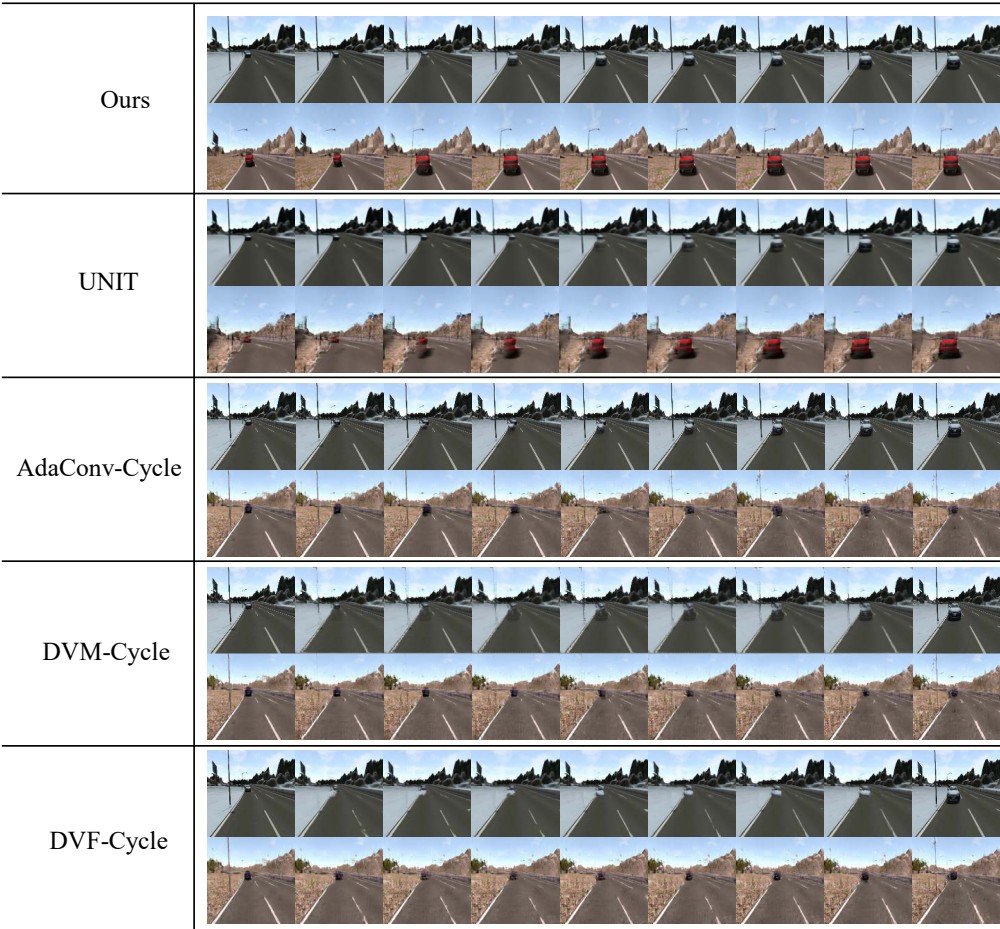

Figure 4: Comparisons of different methods for season winter→spring translation on SYNTHIA dataset. The figure shows all frames of a video synthesized and translated by these mehtods.

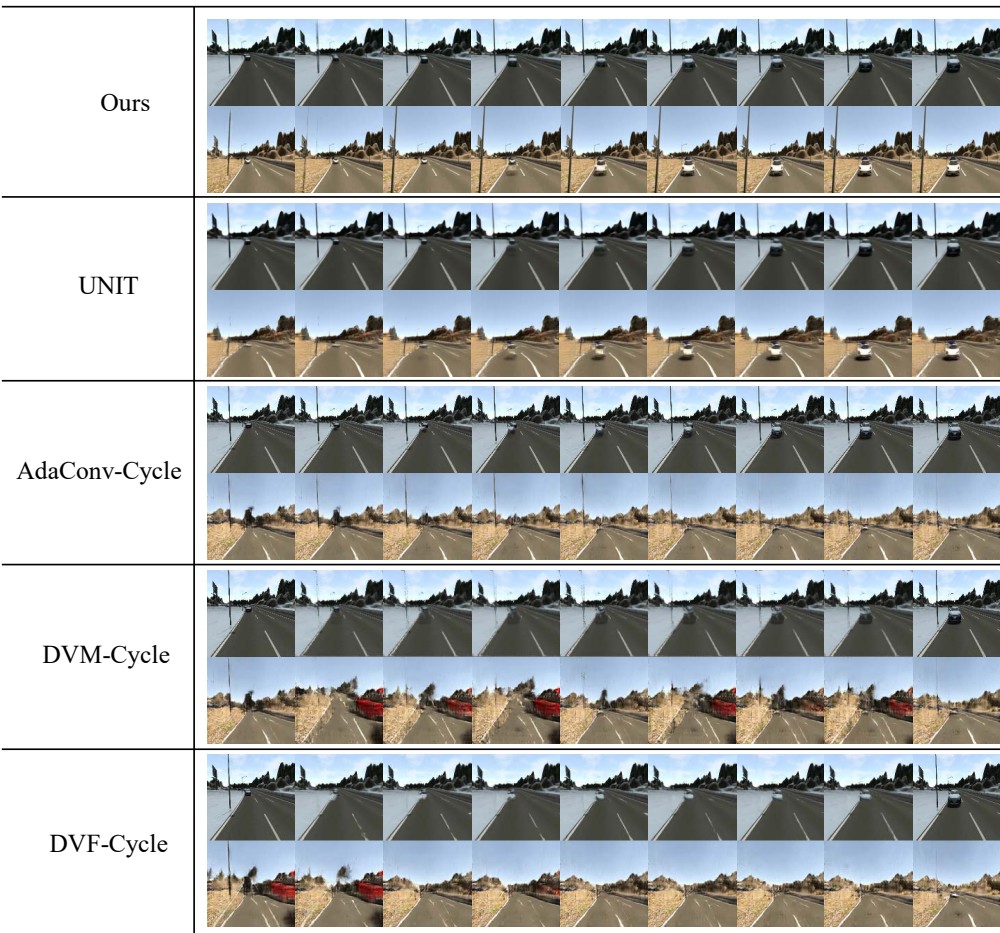

Figure 5: Comparisons of different methods for season winter→summer translation on SYNTHIA dataset. The figure shows all frames of a video synthesized and translated by these mehtods.

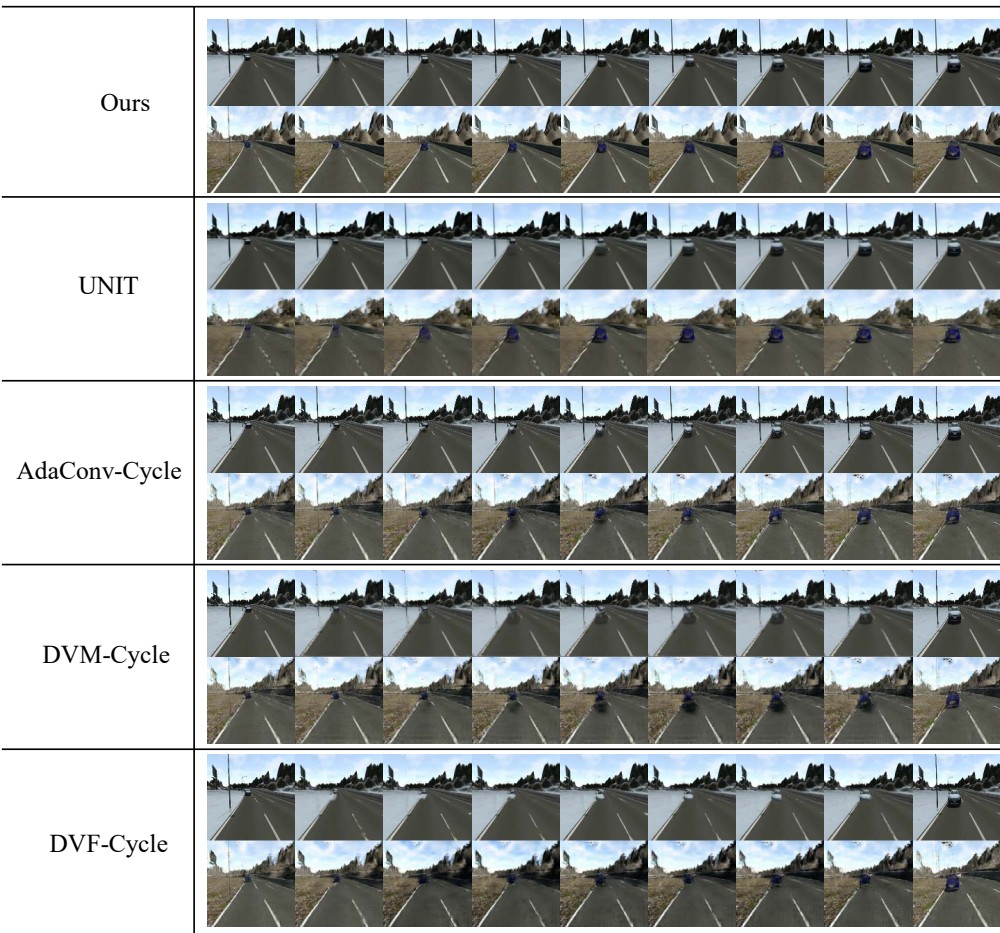

Figure 6: Comparisons of different methods for season winter→fall translation on SYNTHIA dataset. The figure shows all frames of a video synthesized and translated by these mehtods.

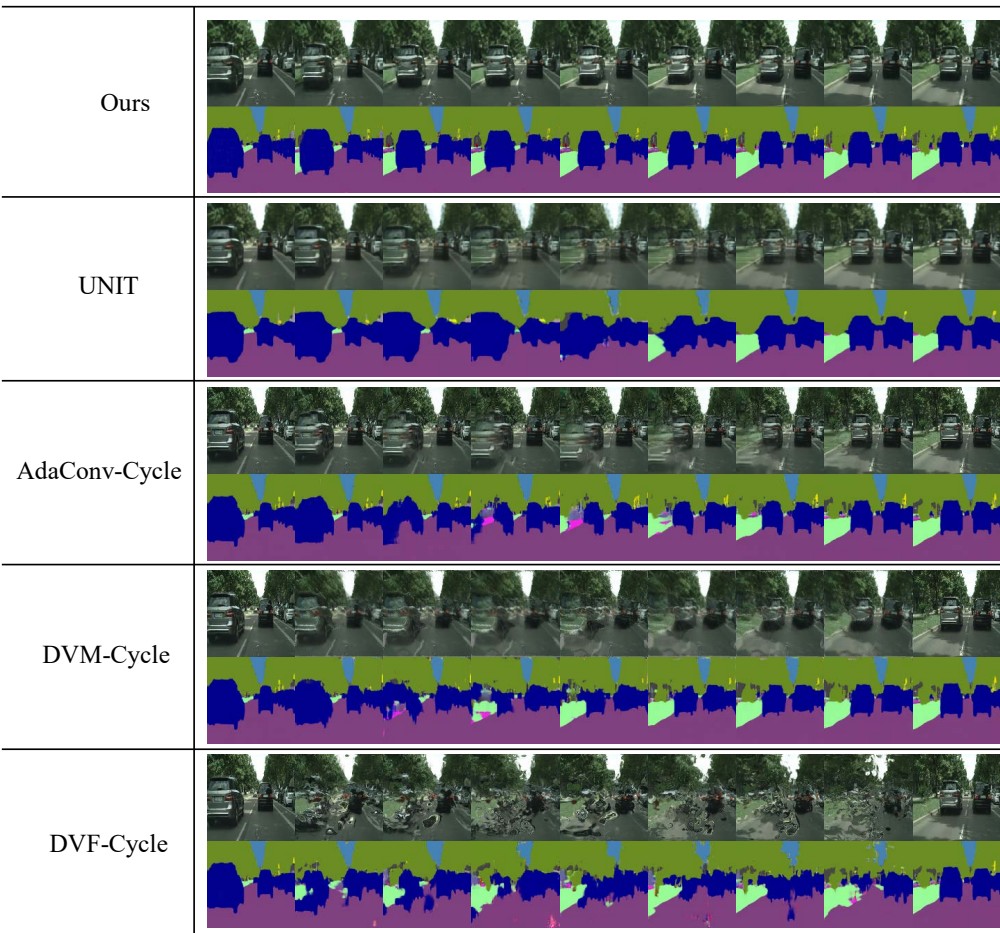

Figure 7: Comparisons of different methods for photo→segmentation translation on Cityscapes dataset. The figure shows all frames of a video synthesized and translated by these mehtods.

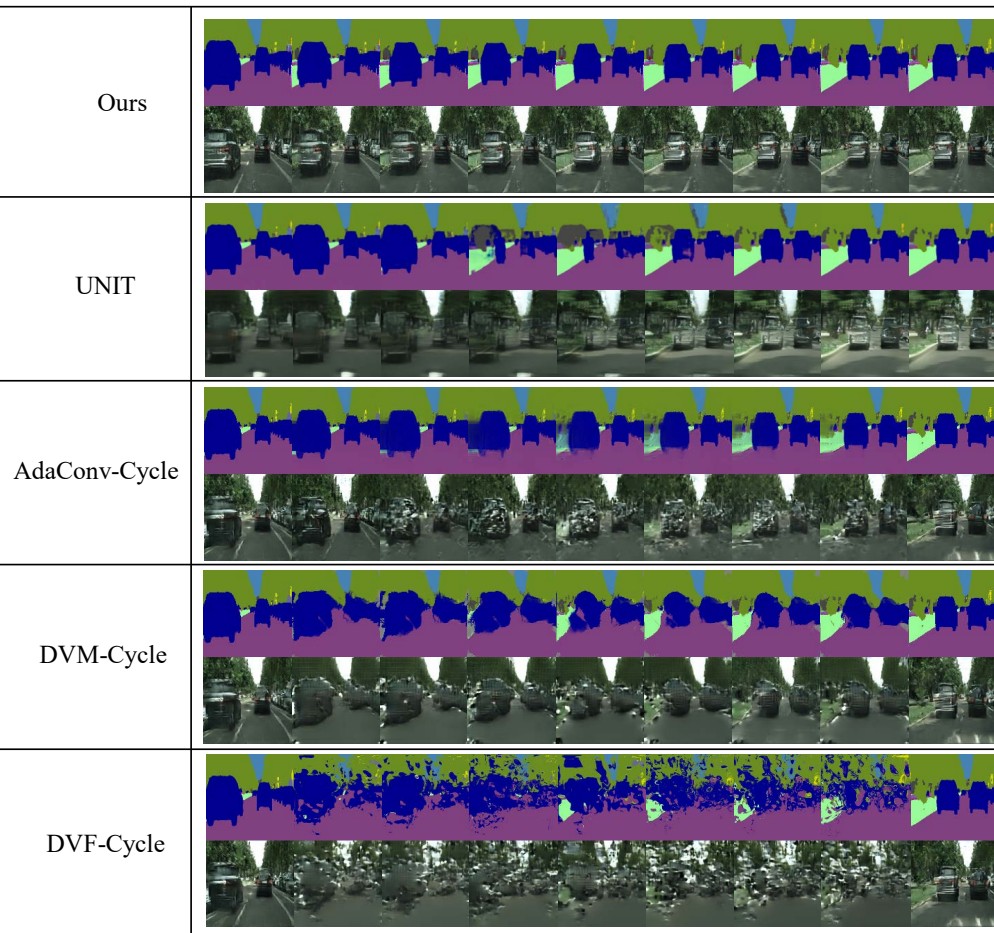

Figure 8: Comparisons of different methods for segmentation→photo translation on Cityscapes dataset. The figure shows all frames of a video synthesized and translated by these mehtods.

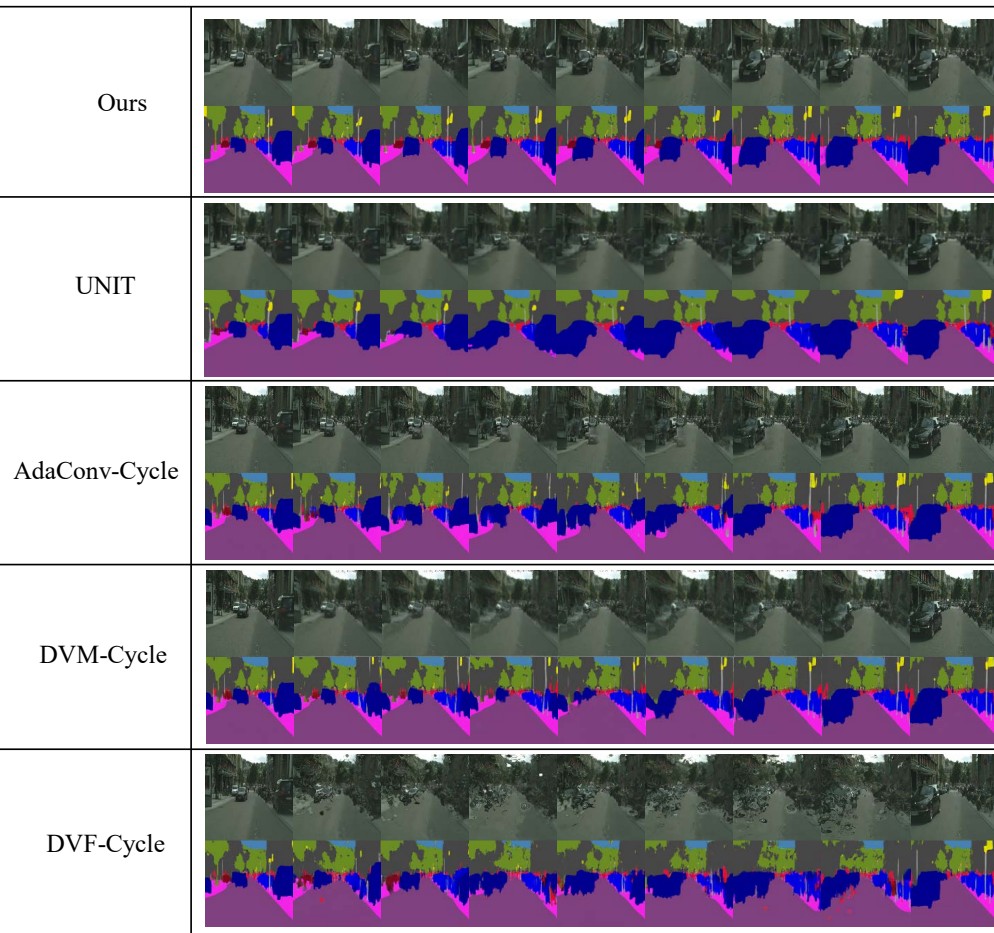

Figure 9: Comparisons of different methods for photo→segmentation translation on Cityscapes dataset. The figure shows all frames of another video synthesized and translated by these mehtods.

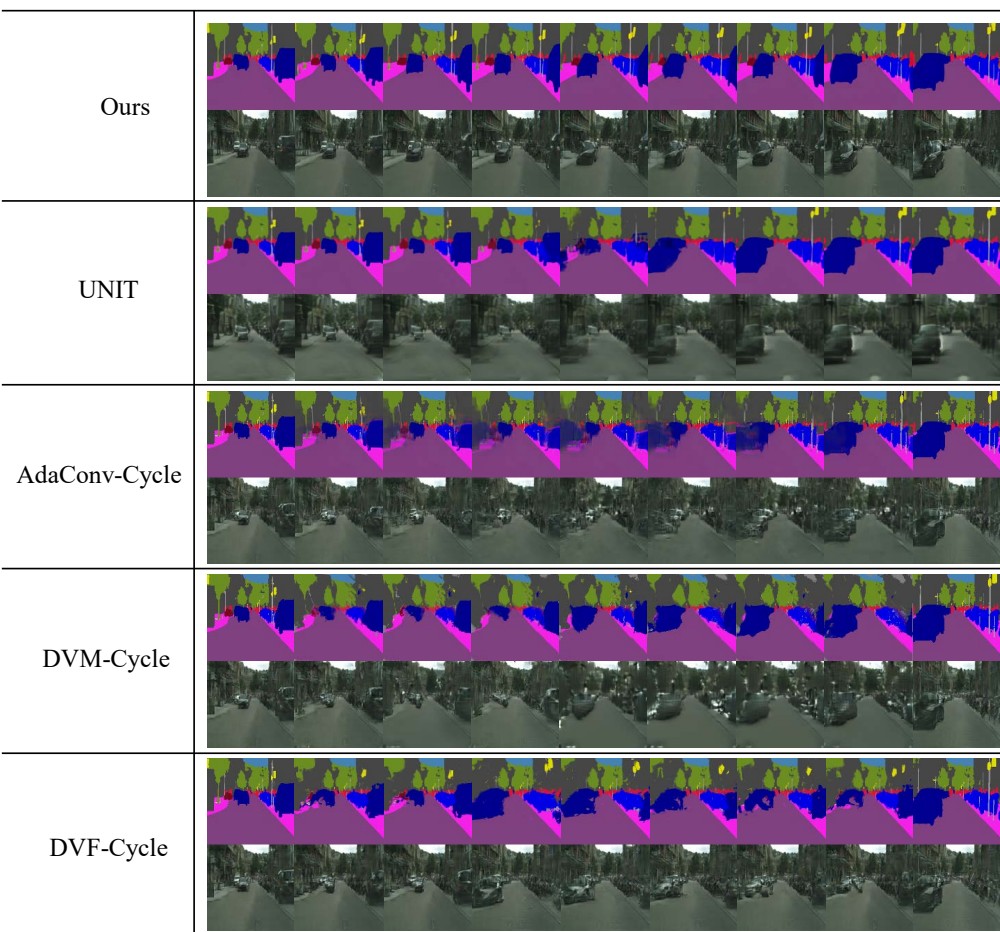

Figure 10: Comparisons of different methods for segmentation→photo translation on Cityscapes dataset. The figure shows all frames of another video synthesized and translated by these mehtods.

## E  TO AR1: CLEAR REVISION HAS BEEN UPDATED

Thanks for your helpful review. We have uploaded a revised version of the paper.

1) We have revised the paper and provided a formal problem definition in Section 3.

2) The definitions of $E_A(f^*)$, $E_B^g(f^*)$ were given in the supplementary material. We have also defined them in Theorem 2 of the revised paper.

3) We have revised the paper and unified notations as follows:
(a) $G2 \circ E1$ and $G1 \circ E2$ denote the cross-domain mappings;
(b) $G1 \circ E1$ and $G2 \circ E2$ represent two Auto-Encoders;
(c) $M$ and $N$ denote the number of samples in the $\mathcal{X}$ and $\mathcal{Y}$ domain, respectively;
(d) $F(s) \lesssim G(s)$ indicates that there exists $C_1, C_2 > 0$ such that $F(s) \leq C_1 G(s) + C_2$.

4) Concerns on $Q_1, Q_2$ and Eqn. (17)
(a) The definitions of $Q_1, Q_2$ in Theorem 1 are reasonable. Taking $Q_1$ as example, the encoder $Q(Z_1|X)$ must satisfy the distribution constraint $Q_{Z_1} = P_Z = Q_{Z_2}, P_Y = P_{G_2}$. Similarly, we can also obtain the constraint for $Q_2$.

(b) The Equality (17) does hold. Firstly, we decompose $P(X, Z_1)$ into $P(X)P(Z_1|X)$. Then, we use $Q(Z_1|X)$ to replace the $P(Z_1|X)$, and enforce its marginal distribution $Q_{Z_1} = E_X[Q(Z_1|X)]$ to be identical to the prior distribution $P_Z$, *i.e.*, $Q_{Z_1} = P_Z$. Meanwhile, the encoder $Q(Z_1|X)$ can be obtained from $E_1(X)$ and $E_2(G_2(E_1(X)))$ when $Q_{Z_1} = Q_{Z_2}$ and $P_Y = P_{G_2}$.

5) Concerns on Lemma 1
According to the duality theorem of Kantorovich-Rubinstein, we can choose any value greater or equal than the considered cost function to satisfy the condition. For simplicity, we choose the equality in the proof of Lemma 1.

6) Formula (24) in Lemma 3 is an inequality.

7) The sets $Q_1$ and $Q_2$ in Problem (4) are different from the definitions in Theorem 1, because the constraints $Q_1$ and $Q_2$ are regularized as penalties. We have made them clearer in the revision.

## F TO AR3: FIRST WORK DERIVED FROM A THEORETICAL PERSPECTIVE

The reviewer undervalued the significance and novelty of the proposed method. We have highlighted them in the revision.

**1) Novelty**
Most image-to-image translation (I2IT) methods (*e.g.*, Cycle GAN) are often designed without a theoretical analysis, which, however, may limit the understanding and the learning performance. Unlike existing methods, to our knowledge, our method is the first work to solve the I2IT problem from a theoretical perspective. Essentially, our method can be regarded as a generalization of CycleGAN. More critically, we believe that our theoretical results would be helpful for understanding the I2IT problem.

**2) Results of image translation**
We have shown the results of image-to-image translation in Table 1 and Figures 2 and 3. Furthermore, we also conducted interpolation-based video-to-video synthesis to evaluate the quality of the learned joint distribution, which is a better method to evaluate the generalization ability of JWAE.

**3) Generative power of the proposed method**
On Cityscapes and SYNTHIA, we compare FID scores of the models trained with different distribution divergences and show the results in Table A. In general, lower FID indicates better performance. From Table A and Figure 11, GAN-JWAE consistently outperforms other distribution divergences. It is worth mentioning that our method is not restricted to the choice of distribution divergences.

Table A: FID results for different distribution divergences

| Method | Cityscapes | | SYNTHIA | | |
|---|---|---|---|---|---|
| | scene2segmentation | segmentation2scene | winter2spring | winter2summer | winter2fall |
| WGAN-JWAE | 134.40 | 84.00 | 122.27 | 97.86 | 86.52 |
| SN-WGAN-JWAE | 128.01 | 87.60 | 117.05 | 97.72 | 85.57 |
| GAN-JWAE | 21.89 | 42.13 | 88.26 | 84.37 | 83.26 |

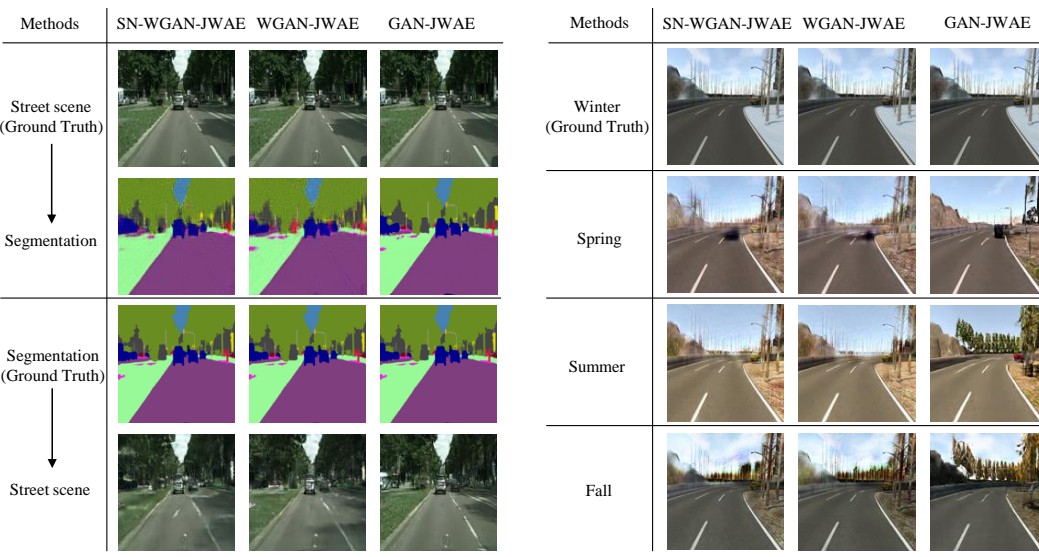

Figure 11: Comparisons of different methods on Cityscapes (left) and SYNTHIA (right).

## G    TO AR2: MISUNDERSTANDING FOR EXPERIMENTS; DIFFICULT EXTENSION FOR I2IT AND V2VS PROBLEM

The reviewer misunderstood the results in Table 1. We have described them clearly in the revision.

**1) & 2) Issues on baselines**
Our method (JWAE) focus on different problem and different experiment settings from Bicycle-GAN, Triple GAN and Triangle GAN. Specifically, JWAE focus on the unsupervised learning setting for image-to-image translation and video-to-video synthesis tasks. In the training, JWAE is trained on unpaired training data. However,
(a) Bicycle-GAN requires paired data and is trained in a supervised setting;
(b) The regularizations in Triple GAN are designed for the classification task and the semi-supervised learning;
(c) Triangle GAN requires the semi-supervised learning. Thus, the comparison between JWAE and these methods is not fair.

**2) Concerns on MMD**
On Cityscapes and SYNTHIA, we show the results using MMD and GAN, respectively. Note that MMD is to measure the distribution divergence of embeddings. Here, we use FID and FID4Video to evaluate the quality of images and videos, respectively. In general, lower score indicates better performance. From Table B, MMD-JWAE can achieve the comparable performance with GAN-JWAE.

Table B: FID results for different distribution divergences

| Method | Cityscapes | | | | SYNTHIA | | | | | |
|---|---|---|---|---|---|---|---|---|---|---|
| | scene2segmentation | | segmentation2scene | | winter2spring | | winter2summer | | winter2fall | |
| | FID | FID4Video | FID | FID4Video | FID | FID4Video | FID | FID4Video | FID | FID4Video |
| MMD-JWAE | 41.60 | 6.62 | 42.65 | 23.48 | 89.72 | 21.18 | 83.10 | 19.88 | 86.56 | 15.99 |
| GAN-JWAE | 22.74 | 6.80 | 43.48 | 25.87 | 88.24 | 21.37 | 77.12 | 17.99 | 87.50 | 14.14 |

**3) Concerns on W/O-Triple**
There is a misunderstanding of the results in Table 1. The baseline method W/O-Triple indicates that we remove the second term (*i.e.*, cycle adversarial (CA) term) in the right-hand side of Loss (6), instead of removing the whole Triple-GAN loss. We refer to this baseline as W/O-CA and we have clarified this in the revision.

**4) Generalization ability of JWAE**
We have shown the generalization ability of JWAE in the paper. According to (Bojanowski et al., 2018), the generalization ability can be evaluated by the interpolation performance in the target domain. From the results in Table 1 and Figures 2 & 3, our method consistently outperforms the considered baseline methods both quantitatively and qualitatively.

**5) Difficulty of applying WAE to I2IT and V2VS tasks**
It is very difficult to extent WAE to image-to-image translation (I2IT) and video-to-video synthesis (V2VS) tasks, because there exists an intractable joint distribution matching problem. Moreover, the original WAE aims at learning a generative model from noise to images, and thus it cannot be directly applied in I2IT and V2VS problem.

