# OpenReview forum: "Learning Joint Wasserstein Auto-Encoders for Joint Distribution Matching"
_ICLR.cc/2019/Conference_

### Official Review · AnonReviewer2 · 2018-11-01
**a straightforward extension to WAE**

**Rating:** 4
**Confidence:** 4

**Review:**

The whole model can be simplified by this: using auto-encoders for X and Y's reconstruction, then use Triple GAN loss to match the joint distribution of (X, Y).  However, the deterministic model with GAN loss looks problematic to me.

questions:

1. Although the authors showed strong evidence in their experiment part, they still failed to compare models with Bicycle-GAN, i.e., how Bicycle GAN performs on these two dataset?

2. missing some comparison: why use simplified Triple-GAN loss (i.e. without two regularization terms)  instead of Triangle-GAN, which is addressed to be better? I think the authors need to discuss about this. Also, the authors need to use MMD and other methods mentioned in the original WAE paper.

3. In table 1, without triple-GAN loss, the whole model is deterministic, but the authors can still show the FID score for the generalization ability, which is better than all other cycle-GAN based models, why is that possible? Is this equivalent to claim that auto-encoder has the ability to generate realistic images just by sampling z?
(If I understand the experiment correctly, the author's synthesized images is generated by $y_hat = G_2(E_1(X))$, no sampling z required)

4. Can the authors show the generalization ability of JWAE? For example, with input X, we can have different correct corresponding Ys, just like Bicycle-GAN did.

---

> ### Author Response · Authors · 2018-11-19
> **To AR2: Misunderstanding for experiments; Difficult extension for I2IT and V2VS problem**
>
> The reviewer misunderstood the results in Table 1. We have described them clearly in the revision.
>
> 1) & 2) Issues on baselines
> Our method (JWAE) focuses on the different problem and different experiment settings from Bicycle-GAN, Triple GAN and Triangle GAN. Specifically, JWAE focuses on the unsupervised learning setting for image-to-image translation and video-to-video synthesis tasks. In the training, JWAE is trained on unpaired training data. However,
> (a) Bicycle-GAN requires paired data and is trained in a supervised setting;
> (b) The regularizations in Triple GAN are designed for the classification task in the semi-supervised learning;
> (c) Triangle GAN requires the semi-supervised learning. Thus, the comparison between JWAE and these methods is not fair.
>
> 2) Concerns on MMD
> On Cityscapes and SYNTHIA, we show the results using MMD and GAN, respectively. Note that MMD is to measure the distribution divergence of embeddings. Here, we use FID and FID4Video to evaluate the quality of images and videos, respectively. In general, lower score indicates better performance. From Table B, MMD-JWAE can achieve the comparable performance with GAN-JWAE.
>
>                               Table B. FID | FID4Video scores on Cityscapes and SYNTHIA
> |   Method      |                 Cityscapes                 |                                    SYNTHIA                                   |
> |                       |  photo2seg    |    seg2photo	| winter2spring | winter2summer |    winter2fall  |
> | MMD-JWAE	 | 41.60 | 6.62   | 42.65 | 23.48  |  89.72 | 21.18  |   83.10 | 19.88     | 86.56 | 15.99 |
> |  GAN-JWAE	 | 22.74 | 6.80   | 43.48 | 25.87	|  88.24 | 21.37  |   77.12 | 17.99     | 87.50 | 14.14 |
>
>
> 3) Concerns on W/O-Triple
> There is a misunderstanding of the results in Table 1. The baseline method W/O-Triple indicates that we remove the second term (i.e., cycle adversarial (CA) term) in the right-hand side of Loss (6), instead of removing the whole Triple-GAN loss. We refer to this baseline as W/O-CA and we have clarified this in the revision.
>
> 4) Generalization ability of JWAE
> We have shown the generalization ability of JWAE in the paper. According to (Bojanowski et al., 2018), the generalization ability can be evaluated by the interpolation performance in the target domain. From the results in Table 1 and Figures 2 & 3, our method consistently outperforms the considered baseline methods both quantitatively and qualitatively.
>
> 5) Difficulty of applying WAE to I2IT and V2VS tasks
> It is very difficult to extent WAE to image-to-image translation (I2IT) and video-to-video synthesis (V2VS) tasks, because there exists an intractable joint distribution matching problem. Moreover, the original WAE aims at learning a generative model from noise to images, and thus it cannot be directly applied in I2IT and V2VS problem.

---

### Official Review · AnonReviewer3 · 2018-11-02
**some issues prevent me from recommending an acceptance**

**Rating:** 5
**Confidence:** 4

**Review:**

This paper proposes a joint Wasserstein Auto-Encoder (JWAE) method to solve the problem of joint distribution matching. Instead of ﬁnding a coupling, the paper seeks a decoupling to make the primal problem of Wasserstein distance tractable. The decoupled version of joint Wasserstein distance is used for empirical reconstruction losses of within-domain Auto-Encodings and cyclic mappings. In addition, two GAN divergences are used to learn the cross-domain mappings such that the generated distributions are close to the real distribution, and another GAN divergence is imposed to align the latent distributions generated by two Auto-Encoders. Later, the paper applies the proposed model on the interpolation based video-to-video synthesis problem.

As far as I understand, the paper can be thought of revisiting the Cycle-Consistent Adversarial Networks (CycleGAN) from the joint Wasserstein Auto-Encoder point of view. In other words, it essentially extends the CycleGAN using additional within-domain auto-encoding reconstruction losses and the latent code alignment loss. Accordingly, the proposed model can be naturally applied to image-to-image translation. I have no idea why the paper merely applies it to interpolation based video-to-video translation. In addition, as the paper tries to apply the relaxed optimal Wasserstein distance to Auto-Encoder and cycle consistency losses, why not apply such Wasserstein distance to the distribution divergence as well. To study the generative power of the proposed generative model using the relaxed Wasserstein distance, it is quite necessary to evaluate the use of exit Wasserstein distance based VAE (e.g., Wasserstein AE) and GAN (e.g., Wasserstein GAN and spectral normalized Wasserstein GAN) losses.

---

> ### Author Response · Authors · 2018-11-19
> **To AR3: First work derived from a theoretical perspective**
>
> The reviewer undervalued the significance and novelty of the proposed method. We have highlighted them in the revision.
>
> 1) Novelty
> Most image-to-image translation (I2IT) methods (e.g., Cycle GAN) are often designed without a theoretical analysis, which, however, may limit the understanding and the learning performance. Unlike existing methods, to our knowledge, our method is the first work to solve the I2IT problem from a theoretical perspective. Essentially, our method can be regarded as a generalization of CycleGAN. More critically, we believe that our theoretical results would be helpful for understanding the I2IT problem.
>
> 2) Results of image translation
> We have shown the results of image-to-image translation in Table 1 and Figures 2 and 3. Furthermore, we also conducted interpolation-based video-to-video synthesis to evaluate the quality of the learned joint distribution, which is a better method to evaluate the generalization ability of JWAE.
>
> 3) Generative power of the proposed method
> On Cityscapes and SYNTHIA, we conduct image translation and compare image FID scores of the models trained with different distribution divergences and show the results in Table A. In general, lower FID indicates better performance. From Table A, GAN-JWAE consistently outperforms other distribution divergences. The qualitative results can be found in Appendix F of the supplementary material. It is worth mentioning that our method is not restricted to the choice of distribution divergences.
>
>                           Table A. FID results for different distribution divergences.
> |       Method        |              Cityscapes          |                                     SYNTHIA                               |
> |                             | scene2seg | seg2scene | winter2spring | winter2summer | winter2fall |
> |    WGAN-JWAE   |    134.40     |      84.00     |         122.27       |           97.86          |      86.52      |
> |SN-WGAN-JWAE |    128.01     |      87.60     |         117.05       |           97.72          |      85.57      |
> |      GAN-JWAE     |      21.89     |      42.13     |           88.26       |           84.37          |      83.26      |

---

### Official Review · AnonReviewer1 · 2018-11-02
**Interesting approach with poor presentation**

**Rating:** 6
**Confidence:** 4

**Review:**

This paper studies the joint distribution matching problem where given data samples in two different domains, one is interested in learning a bi-directional mapping between unpaired data elements in these domains. The paper proposes a joint Wasserstein auto-encoder (JWAE) to solve this problem. The paper shows that under the decomposable cost metric and deterministic decoding maps, the optimization problem associated with the JWAE formulation can be reduced to a tractable optimization problem. The paper also establishes a generalization bound for the JWAE formulation. Finally, the paper conducts an experimental evaluation of the proposed solution with the help of a video-to-video synthesis problem and show improved performance as compared to the existing results in the literature.

Overall, the reviewer finds that the paper considers an important problem and proposes some interesting ideas to tackle the problem. However, in its current form, there is a large scope for improvement in the presentation of the paper. The paper is full of errors/typos which make it an extremely difficult read (see my comments below). That said the paper fairs quite well as compared to other existing methods. Since the reviewer is not very much familiar with this field, the reviewer leaves it to the other reviewers to decide the significance of these results.

Pros:

1) The paper aims to provide a theoretical treatment of the joint distribution matching problem which has many interesting applications, including image-to-image translation and video-to-video synthesis.

2) The proposed method in the paper had good empirical performance on the real world datasets.

Cons:

The paper is very poorly written with many typos and (possibly) mistakes. Some of the comments in this direction are as follows.

1) The paper does not formally define the underlying problem before diving into the details of the proposed solution. The authors only informally talk about the problem in the introduction. Given that the ICLR has a wide audience, it would have been nice if the authors have made the presentation of the paper self-contained.

2) In the same vein, the paper talks about many important quantities without introducing them first. E.g., what are $E_{A}(f^*)$, $E_B^g(f^*)$ etc. in the statement of Theorem 2? These quantities are first defined inside the proofs in the supplementary material!

3) Some of the notation in the paper is also very confusing. For example, cross-domain mapping have two different sets of notations. $(E1oG2, E2oG)$ in Sec. 4.2 and $(G2oE1, G1oE2)$ in Section 5. It should be latter. Similarly, Sec. 4.2 refers to $E1oG1$ and $E2oG2$ as auto-encoders, which should be $G1oE1$ and $G2oE2$, respectively. In Sec. 3, the authors refer to $N$ and $M$ as the number of samples in the $X$ and $Y$ domain, respectively. This is then reversed in Theorem 2 and 4. These are only a small list of large number of such typos. Also, what is the notion defined in the last line of Sec. 3?

4) It is not clear to me why the sets $Q_1$ and $Q_2$ in Theorem 1 are define in their current forms. In particular, it is not clear why the equality hold in Eq. (17) in the proof of Theorem 1.

5) One line in the proof of Lemma 1 says, "Specifically, we choose its equality, then we have". Could the authors elaborate on this?

6) Eq. (24) should be inequality?

7) Given that the authors write a regularized problem in (4). Does that mean now sets $Q_1$ and $Q_2$ are different from how they are defined in the statement of Theorem 1?

#########################

Post rebuttal: The authors have addressed most of my concerns regarding the poor presentation of the earlier version. I have updated my score.

---

> ### Author Response · Authors · 2018-11-19
> **To AR1: Clear revision has been updated**
>
> Thanks for your helpful review. We have uploaded a revised version of the paper.
>
> 1) We have revised the paper and provided a formal problem definition in Section 3.
>
> 2) The definitions of $E_A(f^*)$, $E_B^g(f^*)$ were given in the supplementary material. We have also defined them in Theorem 2 of the revised paper.
>
> 3) We have revised the paper and unified notations as follows:
> (a) $G2 \circ E1$ and $G1 \circ E2$ denote the cross-domain mappings;
> (b) $G1 \circ E1$ and $G2 \circ E2$ represent two Auto-Encoders;
> (c) $M$ and $N$ denote the number of samples in the $X$ and $Y$ domain, respectively;
> (d) $F(s) \lesssim G(s)$ indicates that there exists $C_1, C_2> 0$ such that $F(s) \leq C_1 G(s) + C_2$.
>
> 4) Concerns on $Q_1, Q_2$ and Eqn. (17)
> (a) The definitions of $Q_1, Q_2$ in Theorem 1 are reasonable. Taking $Q_1$ as example, the encoder $Q(Z_1|X)$ must satisfy the distribution constraint $Q_{Z_1} = P_Z = Q_{Z_2}, P_Y = P_{G_2}$. Similarly, we can also obtain the constraint for $Q_2$.
>
> (b) The Equality (17) does hold. Firstly, we decompose $P(X, Z_1)$ into $P(X) P(Z_1|X)$. Then, we use $Q(Z_1|X)$ to replace the $ P(Z_1|X) $, and enforce its marginal distribution $Q_{Z_1} = E_X [Q(Z_1|X)]$ to be identical to the prior distribution $P_Z$, i.e., $Q_{Z_1}=P_Z$. Meanwhile, the encoder $Q(Z_1|X)$ can be obtained from $E_1(X)$ and $E_2(G_2(E_1(X)))$ when $Q_{Z_1} = Q_{Z_2}$ and $P_Y = P_{G_2}$.
>
> 5) Concerns on Lemma 1
> According to the duality theorem of Kantorovich-Rubinstein, we can choose any value greater or equal than the considered cost function to satisfy the condition. For simplicity, we choose the equality in the proof of Lemma 1.
>
> 6) Formula (24) in Lemma 3 is an inequality.
>
> 7) The sets $Q_1$ and $Q_2$ in Problem (4) are different from the definitions in Theorem 1, because the constraints $Q_1$ and $Q_2$ are regularized as penalties. We have made them clearer in the revision.

---

> > ### Comment · AnonReviewer1 · 2018-12-01
> > **Reply to authors**
> >
> > Thank you for addressing my comments.

---

### Meta-Review · Area_Chair1 · 2018-12-14
**Shows promise but requires improvements to presentation to make contribution clear**

**Confidence:** 4
**Recommendation:** Reject

**Metareview:**

This paper proposes a new image to image translation technique, presenting a theoretical extension of Wasserstein GANs to the bidirectional mapping case.

Although the work presents promise, the extent of miscommunication and errors of the original presentation was too great to confidently conclude about the contribution of this work.

The authors have already included extensive edits and comments in response to the reviews to improve the clarity of method, experiments and statement of contribution. We encourage the authors to further incorporate the suggestions and seek to clarify points of confusion from other reviewers and submit a revised version to a future conference.